# $N^6$-methyladenosine of HIV-1 RNA regulates viral infection and HIV-1 Gag protein expression

Nagaraja Tirumuru[1,2†], Boxuan Simen Zhao[3,4,5,6†], Wuxun Lu[1,2], Zhike Lu[3,5,6,4], Chuan He[3,5,6,4*], Li Wu[1,2,7,8*]

[1]Center for Retrovirus Research, The Ohio State University, Columbus, United States; [2]Department of Veterinary Biosciences, The Ohio State University, Columbus, United States; [3]Department of Chemistry, The University of Chicago, Chicago, United States; [4]Department of Biochemistry and Molecular Biology, The University of Chicago, Chicago, United States; [5]Institute for Biophysical Dynamics, The University of Chicago, Chicago, United States; [6]Howard Hughes Medical Institute, The University of Chicago, Chicago, United States; [7]Department of Microbial Infection and Immunity, The Ohio State University, Columbus, United States; [8]Comprehensive Cancer Center, The Ohio State University, Columbus, United States

*For correspondence: chuanhe@uchicago.edu (CH); wu.840@osu.edu (LW)

†These authors contributed equally to this work

Competing interests: The authors declare that no competing interests exist.

**Abstract** The internal $N^6$-methyladenosine (m⁶A) methylation of eukaryotic nuclear RNA controls post-transcriptional gene expression, which is regulated by methyltransferases (writers), demethylases (erasers), and m⁶A-binding proteins (readers) in cells. The YTH domain family proteins (YTHDF1–3) bind to m⁶A-modified cellular RNAs and affect RNA metabolism and processing. Here, we show that YTHDF1–3 proteins recognize m⁶A-modified HIV-1 RNA and inhibit HIV-1 infection in cell lines and primary CD4⁺ T-cells. We further mapped the YTHDF1–3 binding sites in HIV-1 RNA from infected cells. We found that the overexpression of YTHDF proteins in cells inhibited HIV-1 infection mainly by decreasing HIV-1 reverse transcription, while knockdown of YTHDF1–3 in cells had the opposite effects. Moreover, silencing the m⁶A writers decreased HIV-1 Gag protein expression in virus-producing cells, while silencing the m⁶A erasers increased Gag expression. Our findings suggest an important role of m⁶A modification of HIV-1 RNA in viral infection and HIV-1 protein synthesis.

## Introduction

Interactions of host proteins with HIV-1 substantially modulate viral replication and pathogenesis (*Goff, 2007*; *Moir et al., 2011*). Host proteins that interact with HIV-1 nucleic acids or viral proteins can either enhance or inhibit viral replication in cells. The secondary structure model of HIV-1 RNA and its interactions with viral proteins have been recently analyzed (*Kutluay et al., 2014*; *Lavender et al., 2015*; *Watts et al., 2009*; *Wilkinson et al., 2008*); however, it is less clear whether host proteins can post-transcriptionally modify HIV-1 RNA, which may affect interactions between RNA and host or viral proteins, thereby affecting HIV-1 infection.

$N^6$-methyladenosine (m⁶A) is the most prevalent internal messenger RNA (mRNA) modification in eukaryotic organisms and plays pivotal roles in post-transcriptional regulation of gene expression (*Fu et al., 2014*; *Jia et al., 2013*; *Yue et al., 2015*). This methylation is reversible (*Jia et al., 2011*; *Zheng et al., 2013*) and is specifically recognized by a family of reader proteins (*Liu et al., 2014*; *Wang et al., 2015*). The m⁶A modification is widely distributed in mammalian mRNAs, enriched in

**eLife digest** HIV infection is a global health challenge. The antiviral drugs that are currently available limit the ability of the virus to multiply in infected individuals, but they rarely eliminate the virus entirely. A better understanding of how HIV behaves in the cell would help researchers to find a cure for persistent HIV infection. When HIV enters an immune cell, its genetic material – in the form of molecules of ribonucleic acid (RNA) – is used as a template to make molecules of DNA. This viral DNA can integrate into the host cell's DNA, where it is used as a template to make more viral RNA molecules, which are then used to make viral proteins. Some of the viral RNAs are also packaged into new virus particles.

In cells, RNA molecules are often subject to a chemical modification called adenosine methylation, which regulates how that RNA is translated into proteins. Specific enzymes add molecules called methyl tags to particular locations on the RNA, while other enzymes remove them. A family of proteins called YTHDF1–3 recognize and bind to these methyl tags on the RNA and influence how much protein is produced from the target RNA. There is evidence to suggest that the cell can add methyl tags to HIV RNA. However, the extent to which this happens, and what effects this modification has on HIV replication and viral protein production, are not clear.

Tirumuru et al. addressed these questions by analyzing how changing the levels of YTHDF1–3 proteins and the enzymes that add or remove methyl tags in human cells affected HIV infection. The experiments show that YTHDF1–3 proteins inhibited HIV infection in immune cells called T-lymphocytes by recognizing HIV RNA that had been methylated, mainly by targeting the step where the viral RNA is copied into DNA.

Altering the levels of the enzymes that add or remove methyl tags in the cells can change the amount of methyl tags attached to RNA molecules, which alters the amount of HIV protein produced. For example, when more RNA molecules had methyl tags, the cells produced more HIV proteins. These findings suggest that adenosine methylation plays an important role in regulating the ability of HIV to thrive and multiply in T-lymphocytes, which are an important target for HIV. Since the RNAs of other human viruses, such as influenza virus, can also be modified by adenosine methylation, drugs that target this pathway could have the potential to be used to fight a variety of viral illnesses.

the 3' untranslated region (UTR) near the stop codon and also present in the 5' UTR and long exons (*Dominissini et al., 2012*; *Meyer et al., 2012*; *Zhou et al., 2015*). It has been known for almost 40 years that RNAs of influenza virus, adenovirus, Rous sarcoma virus, and simian virus 40 are $m^6A$-methylated (*Beemon and Keith, 1977*; *Canaani et al., 1979*; *Hashimoto and Green, 1976*; *Krug et al., 1976*), although the impact of the $m^6A$ modification on viral replication remains unclear. The recent studies revealed that the $m^6A$ modification of HIV-1 RNA significantly affects viral replication and gene expression (*Kennedy et al., 2016*; *Lichinchi et al., 2016*). Lichinchi *et al.* reported that HIV-1 mRNA contains multiple $m^6A$ modifications and viral infection in a CD4$^+$ T-cell line increases the $m^6A$ levels in both host and viral mRNAs (*Lichinchi et al., 2016*), suggesting a dynamic regulation of $m^6A$ methylomes during HIV-1 infection. In contrast, Kennedy *et al.* only found four clusters of $m^6A$ modifications in the 3' UTR region of the HIV-1 RNA genome that enhance viral gene expression (*Kennedy et al., 2016*).

The dynamic addition, removal, and recognition of $m^6A$ in cellular mRNAs and other types of nuclear RNAs are coordinately regulated by three groups of host proteins, including adenosine methyltransferases (writers), $m^6A$ demethylases (erasers), and $m^6A$-selective-binding proteins (readers) (*Fu et al., 2014*). The methyltransferase complex is composed of METTL3, METTL14 and WTAP (Wilms' tumor 1-associating protein), which add $m^6A$ modification to nuclear RNAs (*Liu et al., 2014*; *Ping et al., 2014*). The RNA demethylases FTO (fat mass and obesity-associated protein) and AlkBH5 (AlkB family member 5) remove the $m^6A$ modification of RNAs (*Fu et al., 2014*; *Jia et al., 2011*). Three host proteins, including YTHDF1, 2, and 3 (YTHDF1–3), have been identified as selective $m^6A$-binding proteins (readers) in mammalian cells (*Dominissini et al., 2012*; *Wang et al., 2014*, *2015*). These $m^6A$-reader proteins preferentially bind methylated mRNAs and control the

stability and translation of target mRNAs (*Dominissini et al., 2012*; *Liu et al., 2014*). Human YTHDF1–3 proteins contain a conserved YTH RNA-binding domain that preferentially binds the m⁶A-containing RNAs and a P/Q/N-rich region that is associated with different RNA-protein complexes (*Fu et al., 2014*). Lichinchi et al. recently reported that silencing of the m⁶A writers or the eraser AlkBH5 decreases or increases HIV-1 replication, respectively (*Lichinchi et al., 2016*). Kennedy et al. showed that m⁶A modifications in the 3' UTR region of HIV-1 RNA enhance viral gene expression by recruiting cellular YTHDF proteins (*Kennedy et al., 2016*). However, neither study examined the m⁶A modification of HIV-1 RNA and its effect on HIV-1 replication in primary CD4⁺ T-cells, nor systemically analyzed the role of the m⁶A writers, erasers, and readers in HIV-1 replication.

Here we show that HIV-1 RNA contains multiple m⁶A modifications enriched in the 5' and 3' UTRs and within several coding genes. We mapped the specific sites in HIV-1 RNA bound by YTHDF proteins in HIV-1-infected cells. We found that overexpression of YTHDF proteins in target cells significantly inhibited HIV-1 infection, while knockdown of these proteins in primary CD4⁺ T-cells enhanced HIV-1 infection. Furthermore, knockdown of the m⁶A writers or the erasers decreased or increased HIV-1 Gag synthesis and virion release in virus-producing cells, respectively. Our findings suggest important functions of the m⁶A reader, writer, and eraser proteins in modulating HIV-1 gene expression and viral infection through the m⁶A modification of HIV-1 RNA.

## Results

### HIV-1 RNA genome contains m⁶A modifications

To investigate the presence of m⁶A in HIV-1 RNA and to map the m⁶A modification within HIV-1 RNA, we isolated RNA samples from CD4⁺ Jurkat T-cells or primary CD4⁺ T-cells infected with replication-competent HIV-1$_{NL4-3,}$ and performed immunoprecipitation (IP) with poly(A)-enriched RNA using m⁶A-specific antibodies, followed by high-throughput RNA sequencing (m⁶A-seq) (*Dominissini et al., 2012*). We identified similar profiles of m⁶A peaks in HIV-1 RNA from these two cell types, which are mainly enriched in the 5' and 3' UTRs as well as the *rev* and *gag* genes of the HIV-1 genome (*Figure 1A,B*). To confirm the m⁶A modification of HIV-1 RNA from virus-producing cells, we transfected HEK293T cells with a plasmid containing full-length HIV-1 proviral DNA (pNL4-3) and extracted total RNA from the transfected cells. Using the same m⁶A-seq approach, we identified multiple m⁶A peaks in HIV-1 RNA, which are enriched in the 5' and 3' UTRs and within overlapped HIV-1 coding genes, such as *tat, rev, env,* and *nef* (*Figure 1—figure supplement 1*). These results confirm the m⁶A modification of HIV-1 RNA despite some differences in m⁶A distributions in HIV-1 infected CD4⁺ T-cells compared to transfected HEK293T cells.

To investigate whether HIV-1 virion RNA contains m⁶A, we isolated HIV-1 RNA from highly purified HIV-1 virions derived from infected CD4⁺ T-cells (*Rossio et al., 1998*; *Wang et al., 2008*), and then performed a quantitative analysis of m⁶A level using liquid chromatography-mass spectrometry (*Jia et al., 2011*). Our data showed that m⁶A in HIV-1 RNA was approximately 0.1% of total adenosines (*Figure 1—figure supplement 2*). Considering 35.8% of HIV-1 genomic RNA (gRNA) (9173 nucleotides) are adenosines (*van Hemert et al., 2014*), our data suggest approximately 3–4 sites of the m⁶A modification in each copy of HIV-1 gRNA, which match the numbers of m⁶A peaks identified by m⁶A-seq (*Figure 1A,B* and *Figure 1—figure supplement 1*). Together, these results confirm that HIV-1 RNA contains m⁶A modifications at multiple sites within the viral genome.

### Distribution of m⁶A in the cellular RNAs and gene ontology (GO) analysis of m⁶A-modified cellular genes

To examine the effect of HIV-1 infection on m⁶A modifications of cellular RNAs, we compared the distribution of m⁶A peaks in cellular RNAs from HIV-1 infected and uninfected T-cells. In Jurkat and primary CD4⁺ T-cells, HIV-1 infection did not significantly affect the percentages of total m⁶A peaks mapped to the human genome in the 5' UTR, coding DNA sequence (CDS), 3' UTR and noncoding regions (*Figure 1—figure supplement 3A,B*). The biological effects of the slightly altered m⁶A topology (<1%) remain to be investigated. To determine whether the preferential m⁶A motif usage in the host cells was altered by HIV-1 infection, we performed consensus sequence analyses within the m⁶A peaks to determine the preferred motifs in cellular RNAs. HIV-1 infection of Jurkat cells or

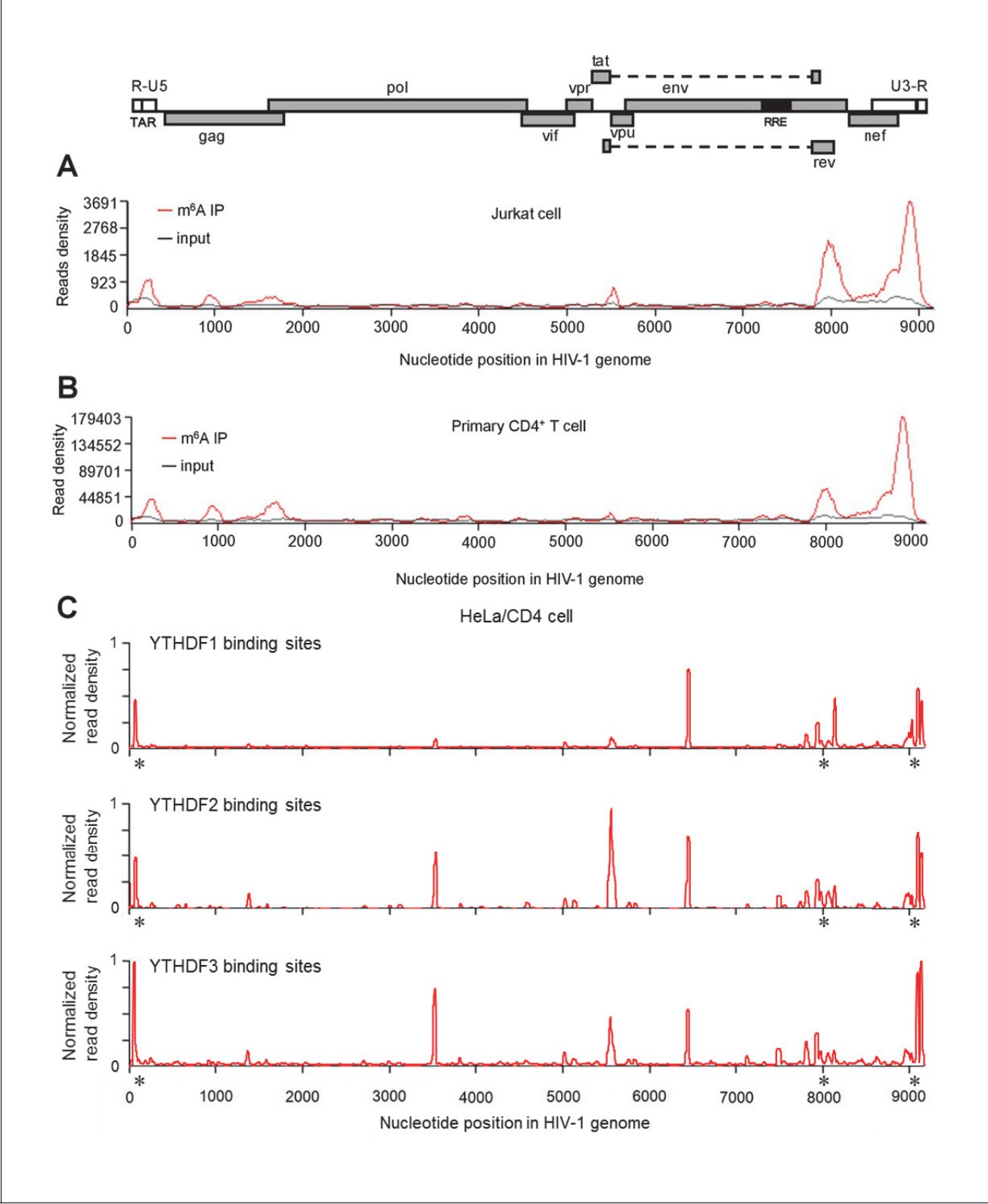

**Figure 1.** HIV-1 RNA contains m⁶A modifications and YTHDF1–3 proteins bind to m⁶A-modified HIV-1 RNA. (**A–B**) The distribution of m⁶A reads from m⁶A-seq mapped to HIV-1 genome (red line) in HIV-1 infected Jurkat cells (**A**) or primary CD4⁺ T-cells (**B**). Baseline signal from the RNA-seq of input samples is shown as a black line. A schematic diagram of HIV-1$_{NL4-3}$ genome is shown above. TAR, transacting response element; RRE, Rev response element. Jurkat cells (**A**) or primary CD4⁺ T-cells (**B**) were infected with HIV-1$_{NL4-3}$ and total RNA was extracted for m⁶A-seq at 72 or 96 hr post-infection

*Figure 1 continued on next page*

Tirumuru *et al*. eLife 2016;5:e15528. DOI: 10.7554/eLife.15528

Figure 1 continued

(hpi), respectively. (C) YTHDF1-3 proteins bind to the HIV-1 gRNA. HeLa/CD4 cells overexpressing FLAG-tagged YTHDF1-3 proteins were infected with HIV-1$_{NL4-3}$ (MOI= 5) for 72 hr and used in CLIP-seq assay to identify their binding sites on HIV-1 gRNA. The distribution of mapped reads (>16 nt) with corresponding nucleotide positions are shown, forming peaks as putative binding positions. Asterisks mark the peak clusters overlapping with identified m$^6$A peaks, indicating high-confident YTHDFs binding sites. Read density was normalized to the total number of mapped reads in each sample (YTHDF1: 28438; YTHDF2: 232568; YHTDF3: 124915). The data presented are representative of results from two independent experiments (n=2).

The following figure supplements are available for figure 1:

Figure supplement 1. HIV-1 RNA contains m$^6$A modifications.

Figure supplement 2. Quantification of HIV-1 RNA m$^6$A level using liquid chromatography-mass spectrometry.

Figure supplement 3. Distribution of m$^6$A in cellular RNAs and the frequency of m$^6$A motifs in HIV-1-infected cells.

Figure supplement 4. Gene ontology (GO) analysis of m$^6$A-modified cellular genes in HIV-1 infected cells.

primary CD4$^+$ T cells slightly increased the frequency of the RRACH motif within the m$^6$A peaks by 0.2–0.8%, but slightly decreased the GGACU motif frequency by 0.2–0.4% (*Figure 1—figure supplement 3C,D*). These data suggest that the preferential usage of the RRACH motifs in m$^6$A modification of cellular RNAs could be altered during HIV-1 infection.

We also performed the GO analysis of m$^6$A-modified cellular genes in HIV-1 infected Jurkat cells and primary CD4$^+$ T cells and found numerous genes with known functions in viral infection-related pathways. We defined these genes as viral-specific genes and performed a separate analysis of the distribution and motif of methylation peaks on these genes (*Figure 1—figure supplement 4A,B*). We have also performed an individual GO analysis of genes with unique methylation peaks in infected samples, the results showed that these genes enrich in functional clusters, such as metabolism, immune system process, multicellular organismal process, and development (*Figure 1—figure supplement 4C,D*), indicating widespread impacts on host biological systems induced by HIV-1 infection.

## YTHDF1–3 proteins bind to HIV-1 gRNA in infected cells

YTHDF1–3 proteins are reader proteins that specifically bind to m$^6$A-methylated cellular RNAs (*Wang et al., 2014*, *2015*). We utilized the crosslinking and immunoprecipitation (CLIP) assay combined with RNA-seq (*Hafner et al., 2010*; *Liu et al., 2015*) to map the binding sites of YTHDF1–3 proteins in the HIV-1 genome in infected HeLa/CD4 cells that overexpressed individual FLAG-tagged YTHDF1–3 proteins. We identified multiple CLIP peaks of YTHDF1–3 protein-bound HIV-1 RNA, including the transactivation response element (TAR) in the 5′ UTR leader sequence, the *env* gene, the *rev* gene, and the 3′ UTR. Some YTHDF-binding sites (e.g. at the 3′ and 5′ UTR and the *gag* gene) in HIV-1$_{NL4-3}$ infected HeLa/CD4 cells partially overlap with the identified m$^6$A-containing regions in the HIV-1 genome in HIV-1$_{NL4-3}$ infected CD4+ Jurkat T-cells or primary CD4+ T-cells (marked by asterisks), indicating high-confident YTHDF1-3 binding sites. Different cell types used in these experiments might contribute to the difference of the m$^6$A sites and YTHDFs-bound sites in HIV-1 RNA genome. Overall, these data demonstrate that YTHDF1-3 proteins bind to m$^6$A-modified HIV-1 genomic RNA during viral infection.

## YTHDF1–3 proteins negatively regulate HIV-1 post-entry infection

Because HIV-1 RNAs are present in the cytoplasm and the nucleus of infected cells at different stages of viral lifecycle (*Goff, 2007*), we hypothesized that YTHDF1–3 proteins may directly interact with methylated HIV-1 RNAs, thereby affecting the metabolism and/or processing of the viral RNA. To examine the roles of YTHDF1–3 proteins in post-entry HIV-1 infection, we either overexpressed or knocked down the individual YTHDF proteins in human cell lines, and examined the effect on HIV-1 infection using a single-cycle, luciferase reporter HIV-1 pseudotyped with vesicular stomatitis virus G protein (VSV-G) to overcome the requirement of HIV-1 receptors during viral entry (*Wang et al., 2007*). Compared to vector control cells, at 24 hr post-infection (hpi), overexpression of individual

FLAG-tagged YTHDF1–3 proteins in HeLa cells (*Figure 2A*) significantly inhibited HIV-1 infection by approximately 10-fold (*Figure 2B*, p<0.0005) and drastically reduced the synthesis of full-length viral Gag protein (Pr55) (*Figure 2C*). In contrast, stable knockdown of individual, endogenous YTHDF1–3 proteins in HeLa cells (*Figure 2D*) significantly increased HIV-1 infection by four- to 14-fold (p<0.05) relative to control cells (*Figure 2E*). Overexpression or knockdown of YTHDF1–3 proteins in HeLa cells did not affect cell proliferation (data not shown).

To confirm these observations in CD4⁺ T-cells, we generated Jurkat cell lines with knockdown of individual, endogenous YTHDF1–3 proteins (*Figure 3A*) and did not observe a significant change in proliferation of the knockdown cells relative to parental or vector-control Jurkat cells (*Figure 3B*). The partial knockdown of YTHDF1 or YTHDF3 in Jurkat cells increased HIV-1 infection by three- to four-fold (p<0.005), while YTHDF2 knockdown slightly increased viral infection (*Figure 3A and C*) at

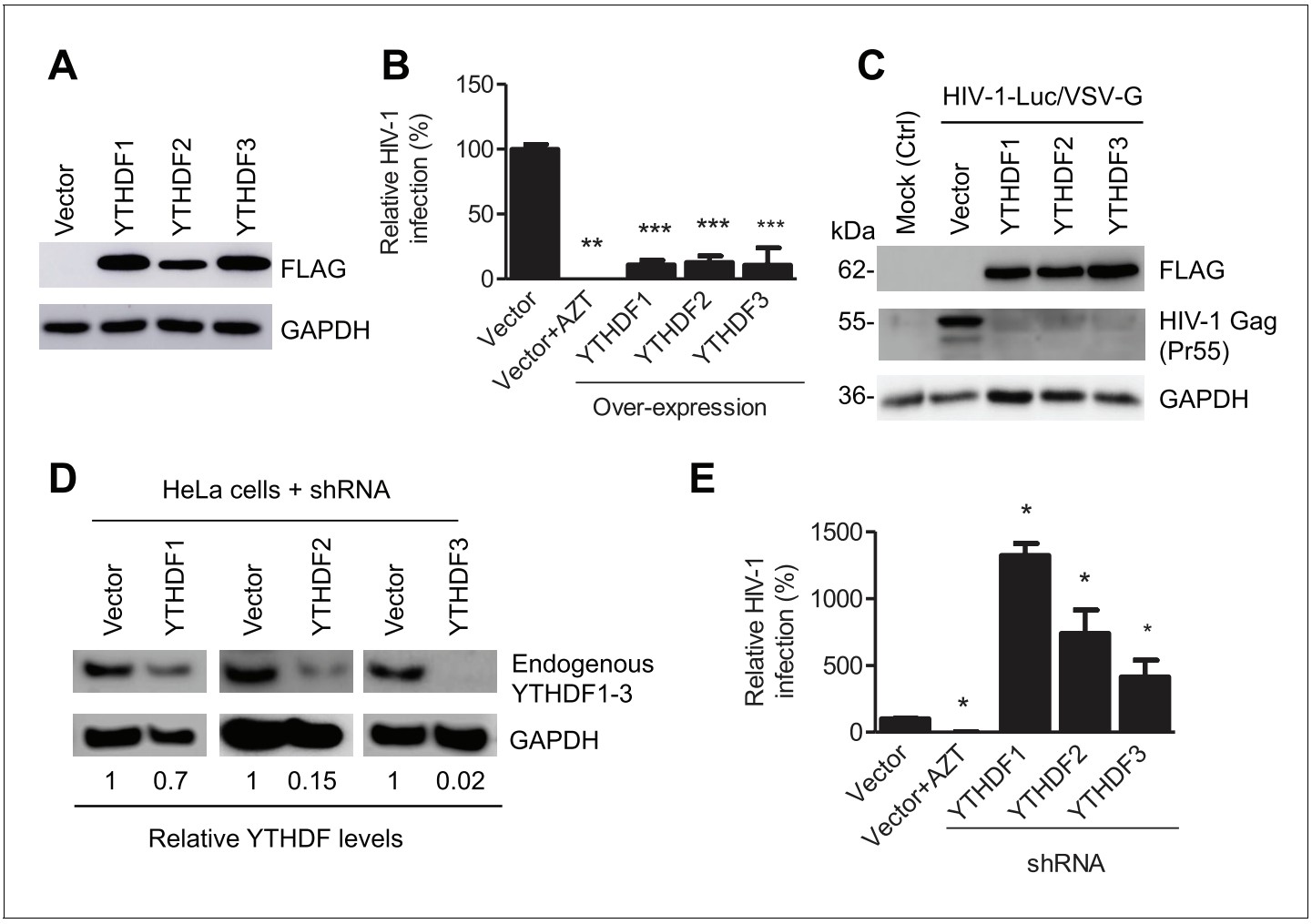

**Figure 2.** YTHDF1–3 proteins negatively regulate post-entry HIV-1 infection in HeLa cells. (A–B) Overexpression of YTHDF1–3 proteins in HeLa cells significantly inhibits HIV-1 infection compared to vector control cells. (A) Overexpression of YTHDF1–3 proteins in HeLa cells was confirmed by immunoblotting. (B) HeLa cells overexpressing YTHDF1–3 proteins were infected with HIV-1 Luc/VSV-G at an MOI of 0.5 and viral infection was measured by luciferase activity at 24 hpi. (C) Overexpression of YTHDF1–3 proteins inhibits HIV-1 Gag protein synthesis in infected cells. HeLa cells overexpressing individual YTHDF1–3 proteins or the vector control cells were infected by HIV-1-Luc/VSV-G at an MOI of 0.5. At 24 hpi, the expression of HIV-1 Gag and YTHDF1–3 proteins (FLAG-tagged) was determined using immunoblotting. GAPDH was used as a loading control and mock-infected vector control cells were used as a negative control. (D and E) Individual knockdown of endogenous YTHDF1–3 proteins in HeLa cells significantly increases HIV-1 infection compared to vector control cells. HIV-1 infection assays were performed as described for panel B. *p<0.05, **p<0.005, and ***p<0.0005, compared to vector control without AZT treatment. All results are shown as mean ± SD (n=3) and data presented are representative of at least three independent experiments.

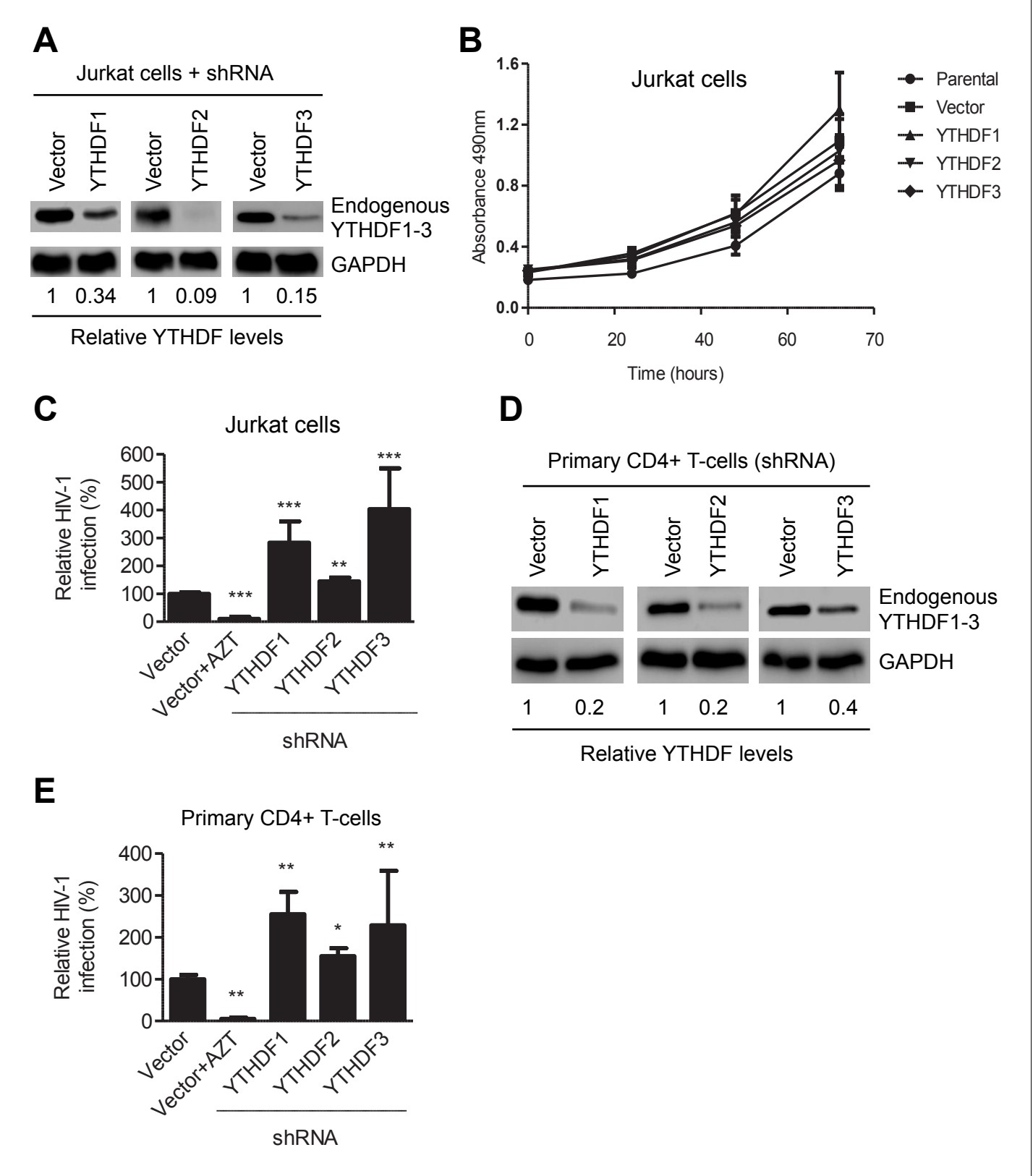

**Figure 3.** YTHDF1–3 proteins negatively regulate post-entry HIV-1 infection in CD4[+] T-cells. (**A**) Individual knockdown of endogenous YTHDF1–3 proteins in Jurkat CD4[+] T cells was confirmed by immunoblotting. (**B**) Knockdown of YTHDF1–3 proteins does not affect proliferation of Jurkat cells. Jurkat cells ($2 \times 10^4$) were seeded and cultured for 3 days. At the times indicated, cell proliferation was measured using the MTS assay. (**C**) Knockdown
*Figure 3 continued on next page*

*Figure 3 continued*

of YTHDF1–3 proteins significantly increases HIV-1 infection compared to vector control cells. (D) Individual knockdown of YTHDF1–3 proteins in activated primary CD4+ T-cells from a healthy donor. (E) Knockdown of YTHDF1–3 proteins significantly increases HIV-1 infection compared to vector control cells. (A and D) GAPDH was used as a loading control. (C and E) The vector controls without AZT were set as 100%. The reverse transcriptase inhibitor AZT treated cells were used as positive control for productive HIV-1 infection. *$p<0.05$, **$p<0.005$, and ***$p<0.0005$, compared to vector control without AZT treatment. All results are shown as mean ± SD (n=3) and data presented are representative of at least three independent experiments.

24 hpi. Furthermore, knockdown of individual, endogenous YTHDF1–3 proteins in activated primary CD4$^+$ T-cells from healthy donors enhanced HIV-1 infection by approximately two-fold ($p<0.005$) (*Figure 3D and E*), confirming the effects observed in cell lines despite a lesser extent. The treatment of cells with the HIV-1 reverse transcriptase inhibitor azidothymidine (AZT) was used as a negative control to show the expected HIV-1 inhibition (*Figures 2B,E*, *3C and E*). Overall, these data suggest that overexpression of YTHDF1–3 proteins significantly inhibits HIV-1 infection, while knockdown of these proteins efficiently promotes HIV-1 gene expression. Thus, endogenous YTHDF1–3 proteins in CD4$^+$ T-cells act as negative regulators to inhibit post-entry HIV-1 infection.

## YTHDF1–3 proteins inhibit HIV-1 infection by blocking viral reverse transcription

To investigate the mechanisms of HIV-1 inhibition by YTHDF1–3 proteins, we assessed the stage of HIV-1 life cycle affected by the YTHDF1–3 proteins. We first measured levels of HIV-1 late reverse transcription (RT) products in infected cells, which represent the levels of the full-length viral cDNA (*St Gelais et al., 2015*). Overexpression of each of the YTHDF1–3 proteins in HeLa cells significantly reduced the level of HIV-1 late RT products by four- to ten-fold ($p<0.005$) compared to the vector control cells at 24 hpi (*Figure 4A*), suggesting that the inhibition of viral reverse transcription contributes to the impairment of post-entry HIV-1 infection. In contrast, the knockdown of individual YTHDF1–3 proteins in HeLa cells elevated the levels of HIV-1 late RT products by two- to three-fold compared to vector control cells (*Figure 4B*). Furthermore, the level of HIV-1 2-LTR circles in infected HeLa cells, a surrogate marker of nuclear import of viral cDNA (*Dong et al., 2007*), was also significantly reduced over 10-fold ($p<0.05$) by overexpression of YTHDF1–3 proteins (*Figure 4C*), corresponding to the reduced late RT products observed in this experiment. Using our established Jurkat cell lines with stable knockdown of individual YTHDF1–3 proteins (*Figure 3A*), we found that the levels of HIV-1 late RT products were significantly increased in YTHDF1 down-regulated cells by 2.7-fold ($p<0.05$) compared to control cells, while the knockdown of YTHDF2 or YTHDF3 only increased late RT products by 20–30% (*Figure 4D*). As a negative control, AZT-treated cells showed inhibition of HIV-1 post-entry infection as expected (*Figure 4A–D*).

The effects on HIV-1 late reverse transcription mediated by YTHDF1–3 would lead to altered viral gene expression. To examine the impacts of the YTHDF proteins on viral gene expression, we quantified HIV-1 *gag* mRNA in infected cells at 24 hpi. The HIV-1 *gag* mRNA level in HeLa cells with overexpression of individual YTHDF1–3 showed a four-fold reduction ($p<0.0005$) compared to vector control cells (*Figure 4—figure supplement 1A*). In contrast, the knockdown of individual YTHDF1–3 in HeLa cells increased the level of *gag* mRNA by eight- to 12-fold ($p<0.05$) compared to vector control cells (*Figure 4—figure supplement 1B*). The knockdown of YTHDF3 in Jurkat cells significantly increased the level of HIV-1 *gag* mRNA by two-fold ($p<0.05$) compared to control cells, while the knockdown of YTHDF1 or YTHDF2 did not have a significant effect (*Figure 4—figure supplement 1C*). The different effects of YTHDF1–3 silencing on *gag* mRNA expression in HeLa and Jurkat cells might result from the difference in the knockdown efficiency in these cells (*Figures 2D* and *3A*). These data suggest that YTHDF1–3 proteins could negatively regulate HIV-1 mRNA transcription, in addition to inhibiting viral reverse transcription in HIV-1 infected cells.

## YTHDF1–3 proteins are associated with HIV-1 gRNA and lead to degradation of viral RNA

We hypothesize that YTHDF1–3 proteins could inhibit the reverse transcription of HIV-1 gRNA through directly binding to the gRNA. To test this hypothesis, we used a single-cycle, VSV-G-

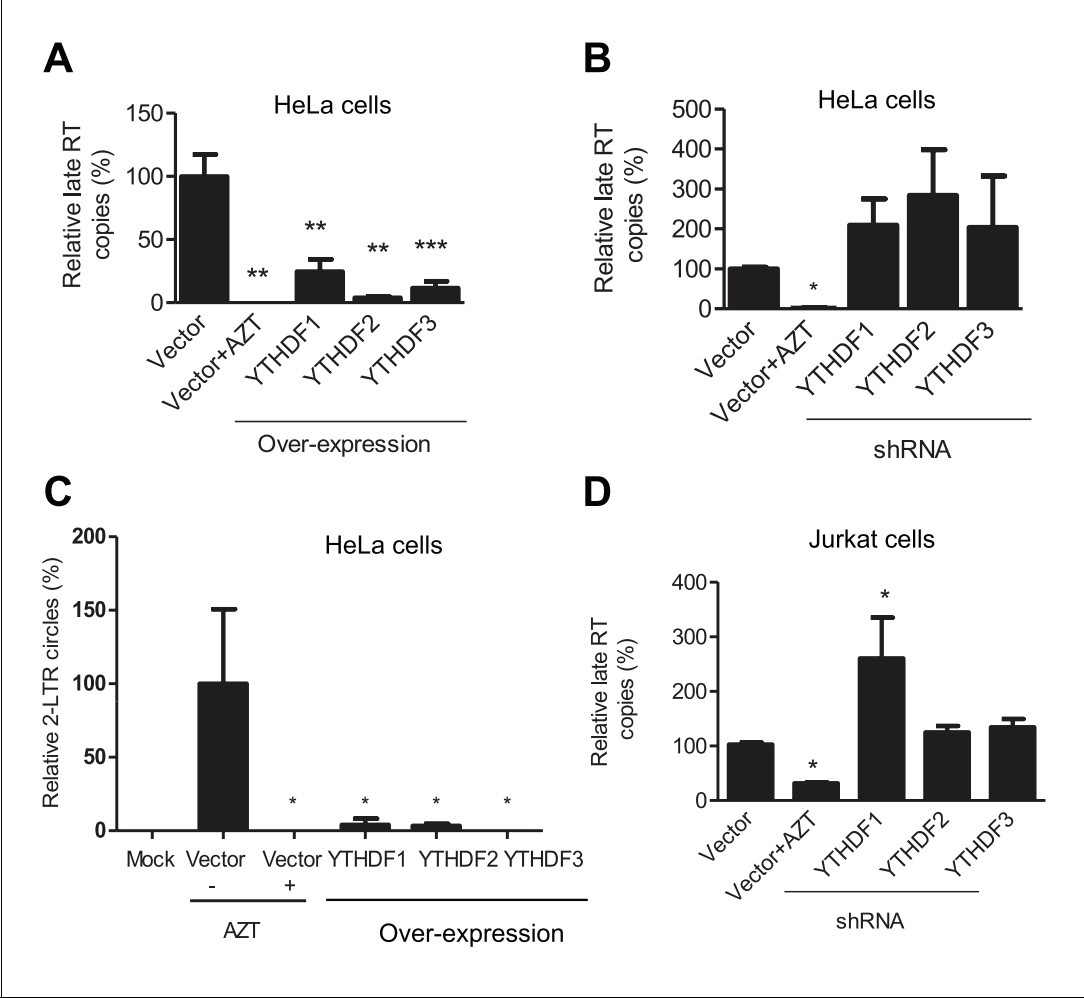

**Figure 4.** YTHDF1–3 proteins inhibit post-entry HIV-1 infection by blocking viral reverse transcription. HeLa cells over-expressing or knocking-down (shRNA) individual YTHDF1–3 proteins were infected with HIV-1-Luc/VSV-G at an MOI of 0.5. (A, B and D) Genomic DNA was isolated from the cells 24 hr post-infection and HIV-1 late reverse transcription (RT) products were quantified by qPCR. (C) YTHDF family proteins reduce the formation of HIV-1 2-LTR circles in infected HeLa cells. At 24 hr post-infection, DNA was isolated from the cells and the 2-LTR circles were analyzed by qPCR and normalized to GAPDH levels. AZT treated vector control cells were used as a negative control for HIV-1 inhibition. *p<0.05 compared to the vector control without AZT treatment. All results are shown as mean ± SD (n=3) and data presented are representative of at least three independent experiments.

The following figure supplement is available for figure 4:

**Figure supplement 1.** YTHDF1–3 proteins negatively regulate HIV-1 *gag* mRNA expression.

pseudotyped HIV-1 to infect HeLa cells overexpressing individual YTHDF1–3 proteins or vector control cells, immunoprecipitated YTHDF proteins from the infected cells at 3 hpi, and then measured HIV-1 gRNA levels in the IP samples. The presence of YTHDF proteins was confirmed in the input and IP samples (*Figure 5A*). The quantification of the HIV-1 gRNA by qRT-PCR revealed a strong and specific association (p<0.005) of HIV-1 gRNA with YTHDF proteins in HIV-1-infected YTHDF1–3-expressing cells compared to control cells (*Figure 5B*). To examine the impact of YTHDF1–3 on HIV-1 *gag* RNA kinetics, we quantified HIV-1 *gag* RNA levels in YTHDF1–3-expressing HeLa cells and vector control cells over a time course of 6–24 hpi. The relative levels of *gag* RNA in HIV-1 infected cells were normalized to that of the vector control cells at 6 hpi. In the control cells, compared to 6 hpi (set as 100%), the level of *gag* RNA was reduced to 40% at 12 hpi and then increased to 80% at 24 hpi (*Figure 5C*), suggesting degradation of HIV-1 gRNA at 12 hpi during the reverse transcription

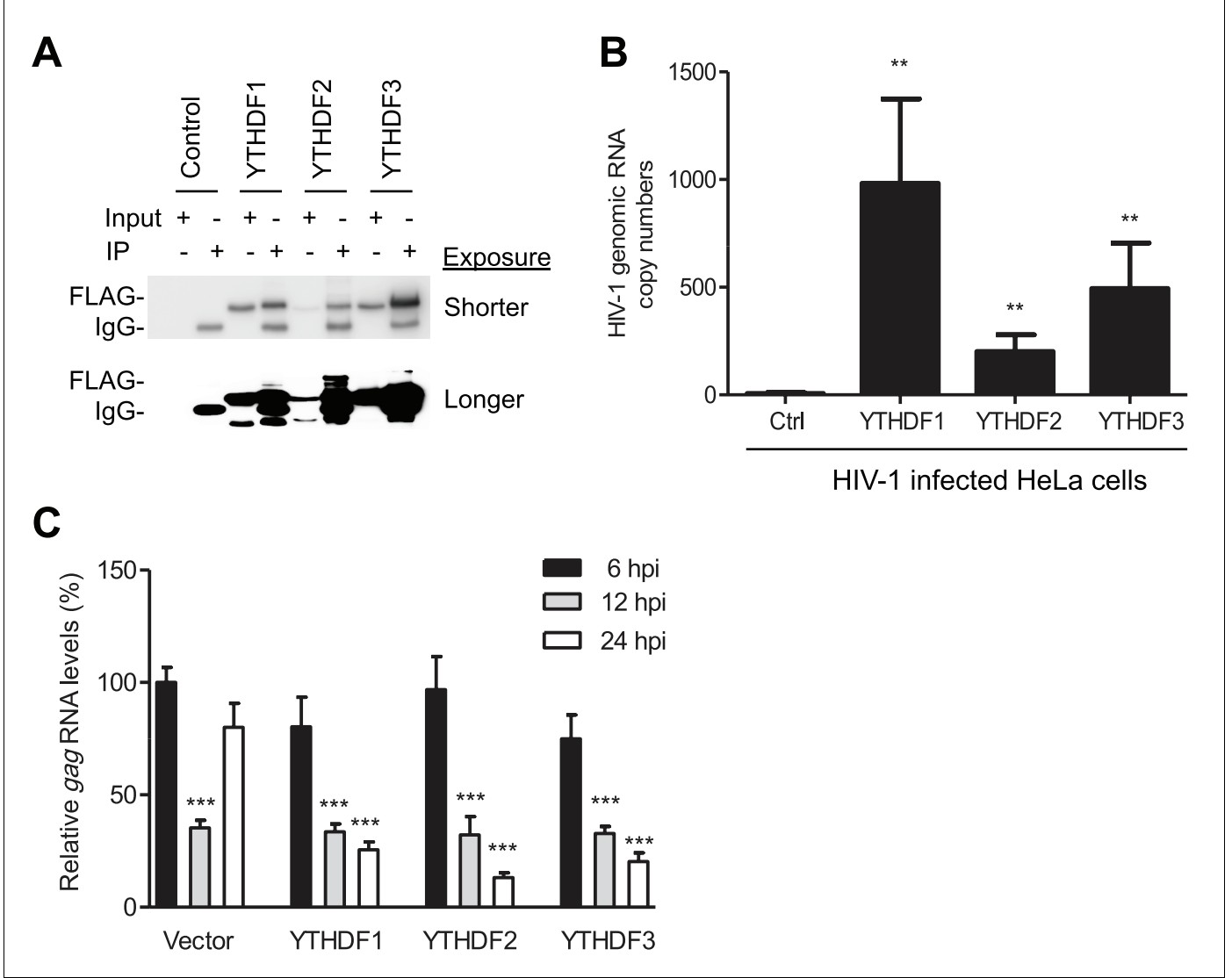

**Figure 5.** YTHDF1–3 proteins bind to HIV-1 gRNA in infected cells. (**A**) Immunoblotting of YTHDF1–3 proteins in the input and immunoprecipitation (IP) samples from HIV-1-Luc/VSV-G infected HeLa cells. FLAG antibodies were used to immunoprecipitate FLAG-tagged YTHDF1–3 proteins overexpressed in HeLa cells after HIV-1 infection. A short and long exposure of the immunoblot is shown. (**B**) HIV-1 gRNA is bound by YTHDF1–3 proteins expressed in HeLa cells. HeLa cells stably overexpressing FLAG-tagged YTHDF1–3 proteins or empty vector control cells (Ctrl) were infected with HIV-1-Luc/VSV-G at an MOI of 5 for 3 hr. Cell lysates were immunoprecipitated with anti-FLAG, RNA was extracted and HIV-1 *gag* RNA levels were measured. **p<0.005 compared to the vector control cells. (**C**) YTHDF1–3 affect HIV-1 *gag* RNA kinetics. HIV-1 *gag* RNA levels in YTHDF1–3-expressing HeLa cells were quantified by qRT-PCR. The relative levels of *gag* RNA in infected cells were normalized to that of the vector control cells at 6 hr post-infection (hpi). ***p<0.0005, compared to the control cells at 6 hpi (set as 100%). All results are shown as mean ± SD (n=3) and data presented are representative of at least three independent experiments.

and then increased *gag* mRNA at 24 hpi during viral gene transcription. In contrast, the levels of *gag* RNA in YTHDF1–3-expressing cells were reduced to 40% at 12 hpi and to 13–25% at 24 hpi (p<0.0005) compared to that of the vector control cells at 6 hpi (*Figure 5C*). These data suggest that YTHDF1–3 proteins can degrade HIV-1 *gag* RNA in infected cells, thereby leading to inhibition of HIV-1 reverse transcription.

# The m⁶A writers and erasers affect HIV-1 Gag expression in virus-producing cells

We examined the role of the m⁶A writers in HIV-1 protein expression and viral release in virus-producing cells. We knocked down endogenous METTL3, METTL14, or both, in HEK293T cells using siRNA, and then transfected the cells with an HIV-1 proviral DNA plasmid (pNL4-3). We determined the levels of HIV-1 Gag protein expression in the cells and the capsid p24 protein released in the supernatants. Interestingly, we found that partial knockdown of METTL3, METTL14, or both, inhibited HIV-1 Gag expression in the cells by 60–70% (*Figure 6A*), and reduced the levels of HIV-1 p24 release by 30–50% compared to control cells (*Figure 6B*). These results suggest that the m⁶A writers are required for efficient HIV-1 protein synthesis, and that m⁶A modification of HIV-1 RNA could facilitate translation of viral proteins.

We next examined the role of the m⁶A erasers in HIV-1 protein expression and viral release in virus-producing cells. We knocked down endogenous AlkBH5, FTO, or both, in HEK293T cells using siRNA, and then transfected the cells with the pNL4-3 plasmid. We determined the levels of HIV-1 Gag protein expression in the cells and the capsid p24 protein released in the supernatants. Interestingly, we found that partial knockdown of FTO significantly promoted HIV-1 Gag synthesis in the cells by 2.5- to 6.5-fold (*Figure 6C*), and increased the levels of HIV-1 p24 release by two- to three-fold compared to control cells (*Figure 6D*). Thus, the m⁶A modification of HIV-1 RNA can enhance HIV-1 protein synthesis. Our results are in agreement with a recent report (*Lichinchi et al., 2016*) showing that silencing of the m⁶A writers (METTL3 and METTL14) or the eraser AlkBH5 decreases or increases HIV-1 p24 expression in the infected MT4 cells, respectively.

## Discussion

The m⁶A modification of cellular mRNAs is coordinately regulated by the writers, erasers, and readers to control the metabolism and processing of methylated RNA (*Fu et al., 2014*). We found that HIV-1 RNA is m⁶A-methylated in infected cells, and that binding of YTHDF1–3 proteins to m⁶A-methylated HIV-1 RNA inhibits viral reverse transcription and gene expression. In contrast, partial knockdown of the m⁶A writers decreased HIV-1 Gag synthesis and viral release, while partial knockdown of FTO had the opposite effects, suggesting that m⁶A modification of HIV-1 RNA could enhance HIV-1 protein synthesis and viral release. Based on our results, we propose a working model suggesting that YTHDF proteins inhibit post-entry HIV-1 infection by blocking viral reverse transcription and mRNA expression, while the m⁶A modification of HIV-1 RNA can promote viral protein translation (*Figure 7*).

It is possible that YTHDF protein-mediated inhibition of HIV-1 infection can result from indirect effects on cellular RNA stability or gene expression, rather than direct inhibition of HIV-1 replication. We noticed a differential level of the effect on HIV-1 infection in different cell types by manipulating the individual YTHDF1–3 proteins. It is possible that the different effects are due to distinct cellular functions and mechanisms of YTHDF proteins (*Fu et al., 2014*). Recent studies indicated that YTHDF1 is responsible for translation promotion, and that YTHDF2 is responsible for mRNA decay, while the function YTHDF3 is unclear, but likely to aid in the temporal-spatial transport and delivery of mRNA (*Wang and He, 2014*; *Wang et al., 2014*, *2015*). Indeed, we observed a unique high peak within the *rev* gene of HIV-1 RNA bound by YTHDF1 (*Figure 1C*), while YTHDF1 and YTHDF2 appear to have similar inhibitory effects on HIV-1 infection. It is possible that YTHDF1 and YTHDF2 may interact with different host proteins that directly or indirectly affect HIV-1 replication and lead to similar effects on viral inhibition. Furthermore, these three YTHDF proteins may have functional redundancy, and individual knockdown of one YTHDF protein may result in a modest effect because the other two could compensate the function. A recent study suggests that the dynamic m⁶A modification of cellular mRNA is a result of stress-induced nuclear localization and upregulation of YTHDF2 (*Zhou et al., 2015*). It is conceivable that HIV-1 infection of cells may induce the changes of cellular localization and expression levels of YTHDF proteins, thereby affecting HIV-1 RNA replication and viral infection.

The mechanisms by which m⁶A modification of HIV-1 RNA regulates viral infection remain to be elucidated. Lichinchi *et al.* recently showed that m⁶A modification of a conserved adenosine (A7883) in the stem loop II region of HIV-1 Rev response element (RRE) RNA increased binding of HIV-1 Rev protein to the RRE and facilitated nuclear export of RNA, thereby enhancing HIV-1 replication

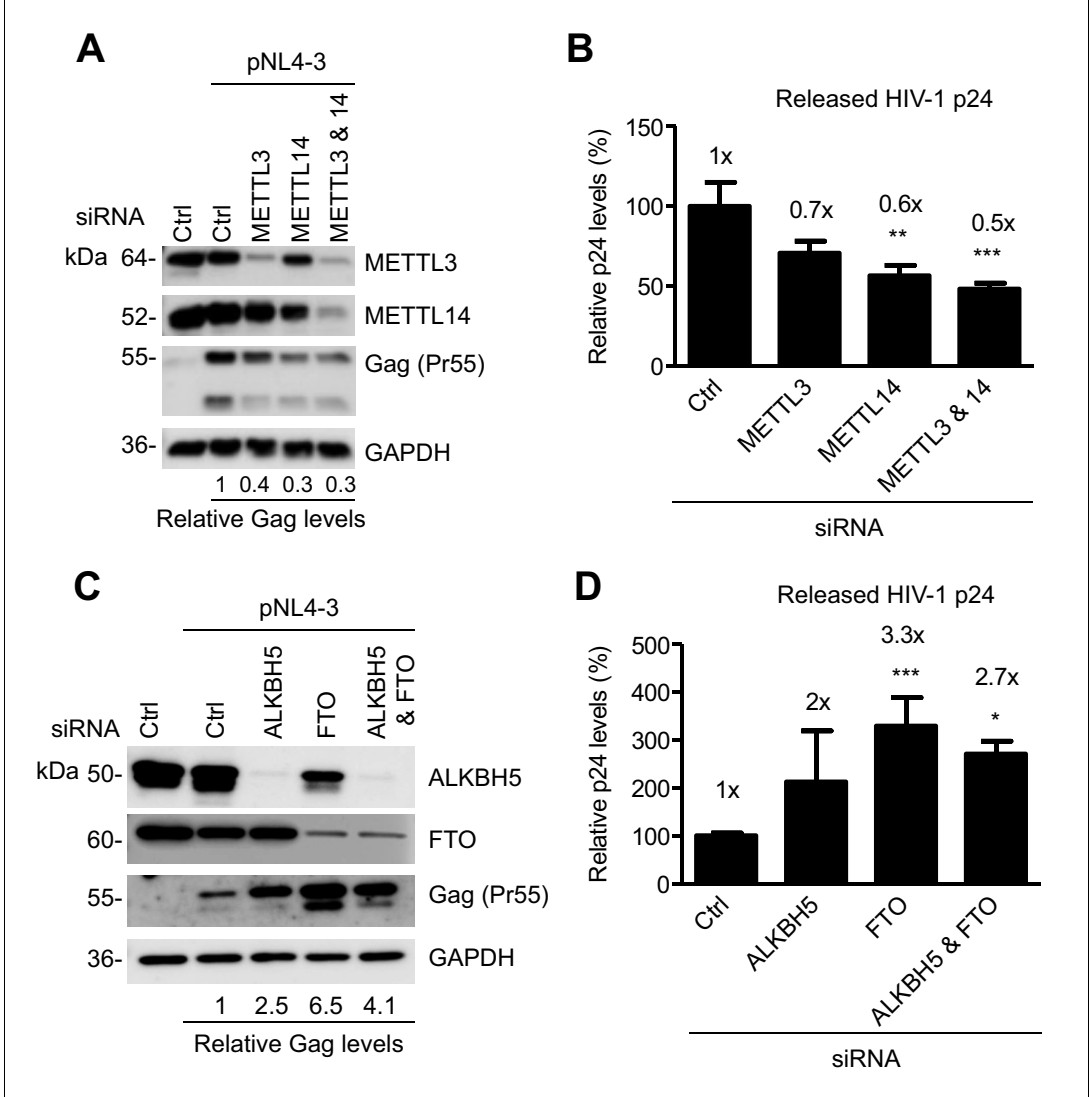

**Figure 6.** The m⁶A writers and erasers affect HIV-1 Gag expression in virus-producing cells. (A and B) Individual or combined knockdown of endogenous METTL3 and METTL14 inhibits HIV-1 Gag protein expression. HEK293T cells were transfected with indicated siRNA, and then with an HIV-1 proviral DNA plasmid (pNL4-3). Cells and supernatants were collected for analyses at 36 hr post-transfection. (A) Expression of METTL3, METTL14 and HIV-1 Gag proteins in the transfected HEK293T cells was detected by immunoblotting. (C and D) Knockdown of endogenous AlkBH5, FTO, or both promotes HIV-1 Gag protein expression. HEK293T cells were transfected with indicated siRNA, and then with pNL4-3. Cells and supernatants were collected at 36 hr post-transfection. (C) Expression of AlkBH5, FTO and HIV-1 Gag proteins in the cells was detected by immunoblotting. (A and C) GAPDH was used as a loading control. Relative levels of Gag expression were normalized to GAPDH levels. (B and D) HIV-1 capsid p24 levels in supernatants were measured by ELISA. The relative levels (%) are also shown. *p<0.05 compared to the siRNA control. The results are shown as mean ± SD (n=3) and data presented are representative of three independent experiments.

(*Lichinchi et al., 2016*). However, other m⁶A sites in the HIV-1 genome (such as in the *gag, pol, tat,* and *rev* genes) might also be critical for viral replication. Future studies using targeted mutations of the identified m⁶A sites in HIV-1 genome may lead to a loss of the effects on HIV-1 infection by YTHDF1–3 knockdown or overexpression.

During the revision of this manuscript, Kennedy *et al.* reported that m⁶A modification of HIV-1 mRNAs enhances viral replication and gene expression (*Kennedy et al., 2016*). Our data of the inhibitory effects on HIV-1 infection by YTHDF1–3 proteins are different from the results recently reported by Kennedy *et al.* (*Kennedy et al., 2016*). They showed that overexpression of YTHDF1–3 proteins in HEK293T cells enhanced HIV-1 mRNA and protein expression in a single-cycle infection, and that

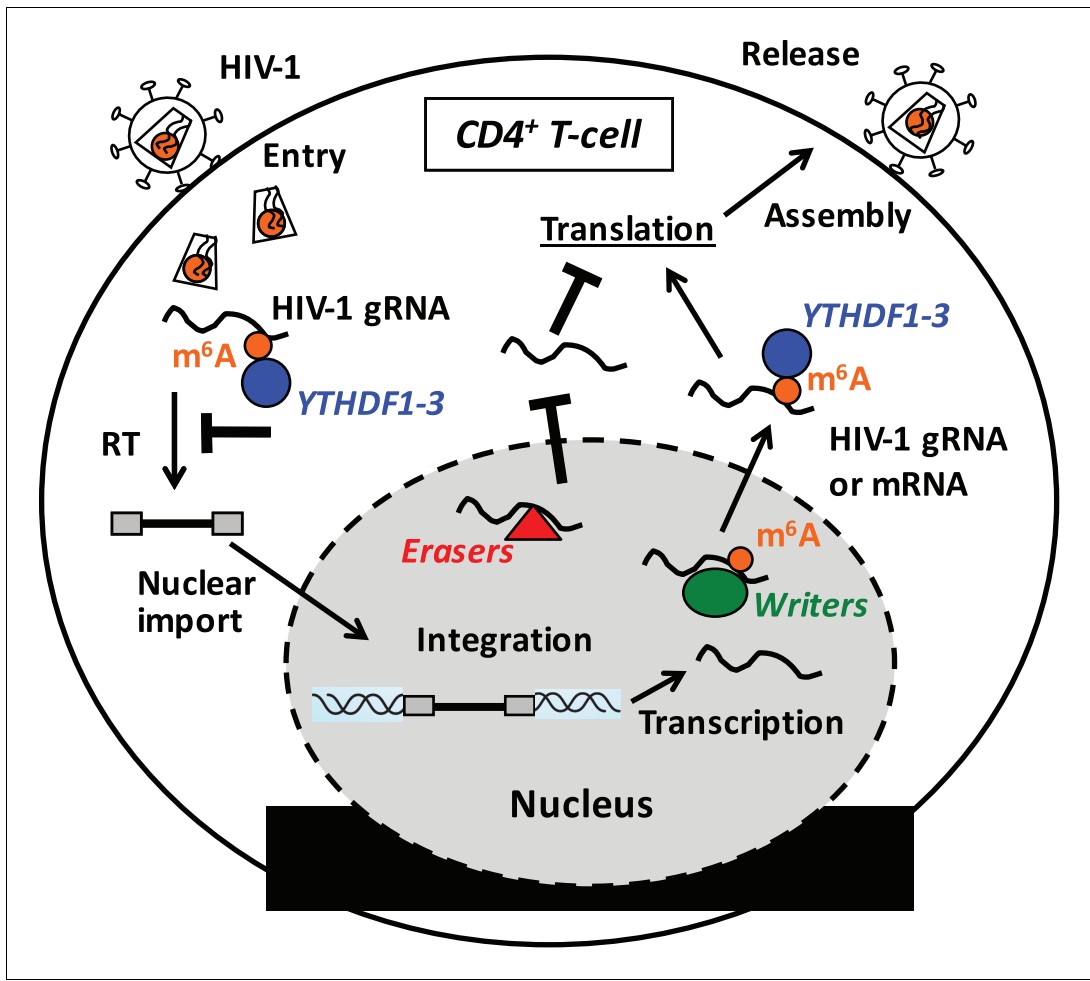

**Figure 7.** Proposed mechanisms and dynamics of m$^6$A modification of HIV-1 RNA in regulating viral infection in cells. In the nucleus, the m$^6$A writers (METTL3 and METTL14) add the m$^6$A marker to HIV-1 genomic RNA (gRNA) or mRNA, and the m$^6$A erasers (FTO and AlkBH5) remove the m$^6$A modifications of HIV-1 RNA. The m$^6$A modification of HIV-1 RNA can promote viral protein translation in cells. In contrast, cytoplasmic m$^6$A readers (YTHDF1–3) bind to m$^6$A-modified HIV-1 gRNA, which can result in inhibition of HIV-1 reverse transcription (RT), viral mRNA expression, and thereby HIV-1 infection in cells.

overexpression or knockout of YTHDF2 in CEM-SS T-cells increased or decreased HIV-1 replication and protein expression, respectively (*Kennedy et al., 2016*). The mapping of m$^6$A sites in HIV-1 RNA between the previous data (*Kennedy et al., 2016*; *Lichinchi et al., 2016*) and the results presented in this manuscript are also different. Different approaches, cell lines and other reagents used in these studies may contribute to distinct results obtained, which remain to be clarified in the future. However, we report here consistent m$^6$A mapping results using Jurkat T-cells and primary CD4$^+$ T-cells and systematic evaluation of the roles of the m$^6$A writers, readers, and erasers in HIV-1 infection.

Chemical modifications of viral RNAs such as m$^6$A may protect viral genomes or mRNA from recognition by cellular innate immunity proteins (*Fu et al., 2014*). The m$^6$A modification of HIV-1 RNA could serve as an immune evasion strategy, wherein the virus may escape detection by innate immunity against infection. In response to HIV-1 infection, the host may evolve and utilize the YTHDF1–3 proteins to bind the m$^6$A-modified viral RNA and inhibit its reverse transcription and subsequent viral mRNA expression. Our findings can stimulate further studies on the precise role of m$^6$A in HIV-1 RNA replication and the mechanisms of m$^6$A regulation pathways that impact HIV-1 infection in cells. Because m$^6$A-modified RNA has also been found in other viruses (*Beemon and Keith, 1977*;

*Canaani et al., 1979*; *Hashimoto and Green, 1976*; *Krug et al., 1976*), this modification pathway may represent a novel and conserved target for antiviral development.

## Materials and methods

### Cell culture

Human healthy primary CD4[+] T-cells were isolated from healthy blood donors' buffy coats (purchased from American Red Cross Blood Service, Columbus, OH) using anti-human CD4-coated magnetic particles according to the manufacturer's instructions (BD Biosciences, San Jose, CA) as described (*St Gelais et al., 2014*, *2015*). Isolated CD4[+] T cells were maintained in complete RPMI media containing interleukin-2 (20 U/ml, PeproTech, Rocky Hill, NJ) and activated with phytohemagglutinin A (PHA, Sigma-Aldrich, St. Louis, MO) as described (*St Gelais et al., 2014*). HEK293T cells, Jurkat cells, and the HIV-1 indicator cell line GHOST/X4/R5 were cultured as described (*St Gelais et al., 2014*, *2015*). HeLa cells overexpressing the empty vector (pPB-CAG), YTHDF1, YTHDF2, or YTHDF3 were maintained in complete DMEM containing 2 μg/ml of puromycin. All parental cell lines were obtained from the American Type Culture Collection (ATCC, Manassas, VA) and the identity of the cell lines has been authenticated using short tandem repeat profiling or genotyping methods as described (*Wu et al., 2004*). All the cell lines were tested negative for mycoplasma contamination using a PCR-based universal mycoplasma detection kit (ATCC).

### Cell proliferation assay

Cell proliferation of HeLa cells or Jurkat cells with YTHDF1–3 overexpression or knockdown respectively were determined by the MTS assay (Promega, Madison, WI) as described (*St Gelais et al., 2015*). Cells ($2 \times 10^4$) were plated in triplicate in a 96-well plate and cultured for 3 days and the absorbance was read at 490 nm at the indicated times.

### Plasmids and shRNA- and siRNA-mediated gene silencing

HIV-1 proviral DNA vector pNL4-3, pNL-Luc-E[−]R[+] containing a firefly luciferase reporter gene and the empty vector control were described (*de Silva et al., 2012*). The pPB-CAG plasmid vector was used to overexpress the YTHDF1–3 proteins in HeLa cells. pLenti vectors carrying specific YTHDF1–3 shRNAs (*Table 1*) were used to knockdown of YTHDF1–3 proteins in different cell types as described (*St Gelais et al., 2015*). Jurkat cells transduced with lentivirus containing shRNAs specific for YTHDF1, YTHDF2, and YTHDF3 were maintained in puromycin (3 μg/ml) containing complete RPMI media. AlkBH5, FTO, METTL3 and METTL14 gene expression in HEK293T cells was silenced by two rounds of siRNA transfection using specific siRNA (Qiagen, Valencia, CA, sequences listed in *Table 2*) transfected with the Lipofectamin RNAiMax reagent (Invitrogen, Waltham, MA) according to the manufacturer protocol (reversible siRNA transfection method). Briefly, HEK293T cells ($1.5 \times 10^5$) were transfected with specific siRNA or a non-specific control (80 nM). At 24 hr post-transfection, media were replaced and the second round of siRNA transfection was performed using the same siRNA concentration (80 nM). The pNL4-3 construct (0.5 μg) was transfected into the cells ($1.5 \times 10^5$) 6 hr after of the second round transfection and cells were harvested for immunoblotting 36 hr after the proviral DNA transfection.

**Table 1.** The shRNA sequences used in this study.

| shRNA | Sequences (5'-3') |
|---|---|
| Non-specific (vector) control | CCGGCAACAAGATGAAGAGCACCAACTCGAGTTGGTGCTCTTCATCTTGTTGTTTTT |
| YTHDF1 | CCGGCCCGAAAGAGTTTGAGTGGAACTCGAGTTCCACTCAAACTCTTTCGGGTTTTTG |
| YTHDF2 | CCGGCGGTCCATTAATAACTATAACCTCGAGGTTATAGTTATTAATGGACCGTTTTTG |
| YTHDF3 | CCGGGATAAGTGGAAGGGCAAATTTCTCGAGAAATTTGCCCTTCCACTTATCTTTTTG |

**Table 2.** The siRNA sequences used in this study.

| siRNA | Sequences (5'-3') |
| --- | --- |
| METTL3 | 5'-CTGCAAGTATGTTCACTATGA-3'<br>5'-AGGAGCCAGCCAAGAAATCAA-3' |
| METTL14 | 5'-TGGTGCCGTGTTAAATAGCAA-3'<br>5'-AAGGATGAGTTAATAGCTAAA-3' |
| FTO | 5'-AAATAGCCGCTGCTTGTGAGA-3' |
| AlkBH5 | 5'-AAACAAGTACTTCTTCGGCGA-3' |

## HIV-1 infection assays

Single-cycle, luciferase reporter HIV-1 stock (HIV-Luc/VSV-G) was generated by calcium phosphate co-transfection of HEK 293T cells with the pNL-Luc-E⁻R⁺ and pVSV-G as described (*St Gelais et al., 2014*). The infectious units of virus stocks were evaluated by limiting dilution on GHOST/X4/R5 cells as described (*Wang et al., 2007*). HIV-1 infection assays using luciferase reporter viruses were performed using a multiplicity of infection (MOI) of 0.5 as described (*de Silva et al., 2012*). Cell lysates were obtained 24 hpi and analyzed for luciferase activity using a commercially available kit (Promega) according to the manufacturer's instructions. Total cell protein was quantified using a bicinchoninic acid assay (BCA; Pierce, Waltham, MA) and all luciferase results were normalized to total protein amounts. HIV-1 capsid p24 levels in supernatants were measured by an enzyme-linked immunosorbent assay (ELISA) using anti-p24-coated plates (AIDS and Cancer Virus Program, NCI-Frederick, MD) as described (*Wang et al., 2007*). Jurkat cells were infected with replication-competent HIV-1$_{NL4-3}$ at an MOI of 0.5 as described (*Wang et al., 2007*). At 72 hpi, cells were washed 3 times and harvested for total RNA extraction using RNAeasy kit (Qiagen). PHA-activated primary CD4⁺ T cells were infected with HIV-1$_{NL4-3}$ (40 ng p24 equivalent HIV-1 per $10^6$ cells) and cells were harvested at 96 hpi for total RNA extraction using RNAeasy kit (Qiagen).

## m⁶A-seq

High-throughput sequencing of HIV-1 methylome was carried out using m⁶A-seq (*Dominissini et al., 2012*) and followed the protocol published previously (*Dominissini et al., 2012*). In brief, total RNA containing HIV-1 RNA was extracted from the cells and purified by poly (dT) beads. Purified polyadenylated RNA was mixed with 2.5 µg of affinity purified anti-m⁶A polyclonal antibody (202003; Synaptic Systems, Goettingen, Germany) in IPP buffer (150 mM NaCl, 0.1% NP-40, 10 mM Tris-HCl, pH 7.4) and incubated for 2 hr at 4°C. RNA was used for library generation with the small RNA sequencing kit (NEB, Ipswich, MA). Sequencing was carried out on Illumina HiSeq 2000 according to the manufacturer's instructions.

## Quantification of HIV-1 RNA m⁶A level using liquid chromatography-mass spectrometry (LC-MS/MS)

HIV-1 gRNA (250 µg) was isolated from highly purified HIV-1$_{MN}$ virions (total p24 capsid 600 µg) (*Rossio et al., 1998*; *Wang et al., 2008*) using an RNeasy Mini kit (Invitrogen), and subjected to quantitative analysis of m⁶A level using LC-MS/MS as described (*Jia et al., 2011*).

## Quantitative PCR and RT-PCR

Quantitative PCR (qPCR) was performed to assess the relative levels of HIV-1 late reverse transcription (RT) products and 2-LTR circles as described (*de Silva et al., 2012*; *Dong et al., 2007*). Reverse transcription PCR (RT-PCR) was used to measure HIV-1 *gag* mRNA as described (*Dong et al., 2007*). To amplify HIV-1 late RT products in cells transduced with pLenti vectors, a different set of primers (LW59 and LW60) were used as described (*St Gelais et al., 2015*). Sequences of PCR primers and probes are listed in *Table 3*. All HIV-1 stocks used for PCR assays were treated with DNaseI (40 U/ml; Ambion, Waltham, MA) prior to infections to avoid plasmid DNA contamination.

**Table 3.** Sequences of PCR primers and probes used in this study.

| Primers | Sequences (5'-3') |
| --- | --- |
| HIV-1 *gag* forward | CTAGAACGATTCGCAGTTAATCCT |
| HIV-1 *gag* reverse | CTATCCTTTGATGCACACAATAGAG |
| Unspliced GAPDH forward | GGGAAGCTCAAGGGAGATAAAATTC |
| Unspliced GAPDH reverse | GTAGTTGAGGTCAATGAAGGGGTC |
| Spliced GAPDH forward | GGAAGGTGAAGGTCGGAGTCAACGG |
| Spliced GAPDH reverse | CTGTTGTCATACTTCTCATGGTTCAC |
| MH531 forward (for HIV-1 late reverse transcription (RT) products) | TGTGTGCCCGTCTGTTGTGT |
| BB reverse (for late RT products) | GGATTAACTGCGAATCGTTC |
| HIV-1 late RT product probe | TCGACGCAGGACTCGGCTTGCT |
| 2-LTR probe | AAGTAGTGTGTGCCCGTCTGTTGTGTGACTC |
| 2-LTR forward | GCCTGGGAGCTCTCTGGCTAA |
| 2-LTR reverse | GCCTTGTGTGTGGTAGATCCA |
| LW59 (forward, alternative for late RT detection in shRNA vector-transduced cells) | GACATAGCAGGAACTACTAGTACCC |
| LW60 (reverse, alternative for late RT detection in shRNA vector-transduced cells) | GGTCCTTGTCTTATGTCCAGAATGC |

## Antibodies and immunoblotting

The antibodies used in this study were: anti-GAPDH (clone 4G5, AbD serotec, Atlanta, GA), anti-FLAG (F-3165, Sigma-Aldrich), anti-FTO (ab124892, Abcam, Cambridge, MA), anti- AlkBH5 (HPA007196, Sigma-Aldrich), anti-METTL3 (15073-1-AP, Proteintech Group, Rosemont, IL), anti-METTL14 (HPA038002, Sigma-Aldrich), anti-YTHDF1 (ab99080; Abcam), anti-YTHDF2 (ABE542, EMD Millipore, Billerica, MA), anti-YTHDF3 (ab103328; Abcam), and anti-HIV-1 Gag (clone #24–2, the NIH AIDS Reagent Program). Cells were harvested and lysed in cell lysis buffer (Cell Signalling, Beverly, MA) supplemented with protease inhibitor cocktails (Sigma-Aldrich). Immunoblotting was performed as described (*St Gelais et al., 2015*). Detection of GAPDH (glyceraldehyde-3-phosphate dehydrogenase) expression was used as a loading control.

## Immunoprecipitation and RNA isolation

HeLa cells expressing pPB-CAG vector or YTHDF1–3 ($2.5 \times 10^6$ cells) were seeded in a 60 mm-diameter culture plate. Cells were infected with HIV-Luc/VSV-G at an MOI of 5 for 3 hr. Cells were UV-cross-linked, lysed in cell lysis buffer (Sigma-Aldrich). The cells were incubated in lysis buffer for 10 min on ice with frequent mixing and were sonicated to ensure maximum lysis. The lysed cell suspension was centrifuged for 5 min at 9,300 ×g at 4°C. The supernatant was transferred to fresh tubes and equal amount of cell lysates were mixed with anti-FLAG-conjugated protein G beads and rotated for 2 hr at 4°C. After the incubation, beads were washed 3 times with cell lysis buffer. Co-immunoprecipitated RNA was isolated from the immunoprecipitates using Trizol (Invitrogen), and RNeasy columns (Qiagen) with an on-column DNase I treatment (Qiagen) and eluted with RNase-free water. Equal volumes of RNA were used as a template for first-strand cDNA synthesis, according to the manufacturer's instructions.

## CLIP-seq

We followed a previously reported protocol of the PAR (photoactivatable ribonucleoside-enhanced)-CLIP assay (*Hafner et al., 2010*) with the following modifications. As HIV-1 infection was inhibited by the addition of 4-thiouridine (data not shown), we omitted that step and performed crosslinking directly. Briefly, HeLa/CD4 cells stably expressing individual YTHDF1-3 proteins were seeded in thirty 15-cm diameter plates one day before HIV-1$_{NL4-3}$ infection. At day 2, the cells were infected with HIV-1$_{NL4-3}$ at a multiplicity of infection (MOI) of 5 and cells were washed to remove cell-free viruses. At 72 hr post-infection, the cells were washed once with 10 mL ice-cold PBS. Uncovered cell plates were placed on a tray with ice and irradiated with 0.15 J/cm² of 254 nm UV light three times in a

Stratalinker 2400 (Stratagene, Santa Clara, CA). Cells were scraped off in PBS and transferred to centrifugation tubes and collected by centrifugation at 500 × g for 5 min at 4°C. The cell pellets were lysed in 3 volumes of 1% NP40 lysis buffer and incubated on ice for 10 min. The cell lysates were cleared by centrifugation at 13,000 × g for 15 min at 4°C. Cleared cell lysates were incubated with RNase T1 to a final concentration of 0.2 U/µl, at 22°C for 15 min, and immediately put on ice for 5 min to quench. Samples were then centrifuged at 13,000 × g for 10 min at 4°C and the supernatant was taken. Anti-FLAG M2 magnetic beads (Sigma M8823, 20 µl slurry/15-cm diameter plate) were washed in IP buffer (50 mM HEPES (pH 7.5), 0.3 M KCl, 0.05% NP40) for 5 times and resuspended in cleared cell lysates and incubated at 4°C overhead rotator for 2 hr. After 2 hr, beads were washed with 1 mL IP buffer for 3 times and beads were changed into a new tube for the final wash and resuspended into 100 µl IP buffer. RNAse T1 was added to the beads at a final concentration of 10 U/µl, incubated at 22°C for 6 min, immediately put on ice for 5 min to quench. Beads were washed with 1 mL high salt buffer (50 mM Tris (pH 7.5), 500 mM KCl, 0.05% NP40) five times, then with 1 mL T4 polynucleotide kinase (PNK) buffer containing 50 mM Tris (pH 7.5), 10 mM $MgCl_2$, 50 mM NaCl (without DTT) two times, finally resuspended into 100 µl PNK reaction mix (95 µl commercial PNK buffer and 5 µl T4 PNK (Promega) and incubated at 37°C for 15 min. Then 10 µl PNK and 1.1 µl 10 mM ATP (to a final concentration of 100 µM) were added to the mixture and kept at 37°C for another 20 min, followed by washing with 1 mL PNK buffer (without DTT) five times and with 1 mL high salt buffer five times. The bound RNA fragments were eluted from the beads by proteinase K digestion twice at 55 °C for 20 and 10 min, respectively. The eluate was further purified using RNA Clean and Concentrator kit (Zymo Research). RNA was used for library generation with NEBNext Small RNA Library Prep kit (NEB). Sequencing was carried out on Illumina HiSeq 4000 according to the manufacturer's instructions.

For bioinformatics analysis, after removing the adapter sequence, only reads that are longer than 16 bp were kept. The reads were mapped to the reference genomes (both human hg38 and HIV-$1_{NL4-3}$ (GenBank: M19921.2) using Bowtie2. Reads uniquely mapped to HIV-1 genome (not to human genome) were used in the subsequent analysis.

## Measurement of the kinetics of HIV-1 RNA in infected cells

HeLa cells over-expressing individual YTHDF1–3 proteins or the pPB-CAG vector were infected with HIV-1-Luc/VSV-G (MOI of 0.5). Cells were collected at 6, 12 and 24 hpi. Total RNA was isolated from the cells using RNeasy columns (Qiagen) with on-column DNase I treatment (Qiagen) and eluted with RNase-free water. Quantitative RT-PCR was used to measure HIV-1 *gag* RNA levels as described (*Dong et al., 2007*). Sequences of PCR primers are listed in *Table 3*. All HIV-1 stocks used for infection were treated with DNaseI (40 U/ml; Ambion) prior to infections to avoid plasmid DNA contamination.

## GO analysis

GO analysis was performed using the GO Enrichment Analysis tool from the Gene Ontology Consortium (*Gene Ontology, 2015*). GO graphs were plotted using the Web server REVIGO (*Supek et al., 2011*).

## Statistical analysis

Data were analyzed using Mann-Whitney's U test or one-way ANOVA test with Prism software and statistical significance was defined as $p < 0.05$.

## Data deposition and access

Data accession: all the raw data and processed files have been deposited in the Gene Expression Omnibus (http://www.ncbi.nlm.nih.gov/geo) and accessible under GSE85724.

## Acknowledgements

The authors thank Dr. Jeffrey D Lifson (NCI-Fredrick) for providing purified HIV-1 virions, Suresh de Silva for technical assistance, and Corine St. Gelais for critical reading the manuscript. The antibody to HIV-1 Gag (clone #24-2) and the HIV-1 RT inhibitor AZT were obtained from the NIH AIDS

Reagent Program. This work was supported by NIH grants AI104483 to LW and AI074658 to CH and a seed grant to LW from the Center for RNA Biology at The Ohio State University (OSU). LW is also supported in part by NIH grants (CA181997 and AI120209) and the Public Health Preparedness for Infectious Diseases Program of OSU. CH is an investigator of the Howard Hughes Medical Institute (HHMI). BSZ is an HHMI International Student Research Fellow.

## Additional information

### Funding

| Funder | Grant reference number | Author |
|---|---|---|
| National Institutes of Health | AI074658 | Chuan He |
| Howard Hughes Medical Institute | | Chuan He |
| National Institutes of Health | AI104483 | Li Wu |
| National Institutes of Health | CA181997 | Li Wu |
| National Institutes of Health | AI120209 | Li Wu |
| Center for RNA Biology at The Ohio State University | Seed Grant | Li Wu |

The funders had no role in study design, data collection and interpretation, or the decision to submit the work for publication.

### Author contributions

NT, BSZ, WL, Acquisition of data, Analysis and interpretation of data, Drafting or revising the article; ZL, Acquisition of data, Analysis and interpretation of data; CH, Conception and design, Analysis and interpretation of data, Drafting or revising the article; LW, Conception and design, Acquisition of data, Analysis and interpretation of data, Drafting or revising the article

### Author ORCIDs

Li Wu, http://orcid.org/0000-0002-5468-2487

## Additional files

### Major datasets

The following dataset was generated:

| Author(s) | Year | Dataset title | Dataset URL | Database, license, and accessibility information |
|---|---|---|---|---|
| Nagaraja Tirumuru, Boxuan Simen Zhao, Wuxun Lu, Zhike Lu, Chuan He, Li Wu | 2017 | RNA sequencing dataset of m6A sites and YTHDF protein binding sites in HIV-1 RNA | http://www.ncbi.nlm.nih.gov/geo/query/acc.cgi?acc=GSE85724 | Publicly available at the NCBI Gene Expression Omnibus (accession no. GSE85724) |

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
