## [Decision Letter]

Thank you for submitting your article "N6-methyladenosine of HIV-1 RNA Regulates Viral Infection and HIV-1 Gag Protein Expression" for consideration by *eLife*. Your article has been reviewed by three peer reviewers, one of whom is a member of our Board of Reviewing Editors, and the evaluation has been overseen by Wenhui Li as the Senior Editor. The following individuals involved in review of your submission have agreed to reveal their identity: Kathleen Collins (Reviewer #2).

The reviewers have discussed the reviews with one another and the Reviewing Editor has drafted this decision to help you prepare a revised submission.

The reviewers were collectively in agreement that the topic of RNA m^6^A modification is of much current interest, and that there are interesting observations on its role in HIV-1 replication in the paper. But there were significant weaknesses in the data and the presentation, as should be apparent from the reviews. Some of the findings are confusing or hard to understand. Major revisions are required to answer the criticisms.

Of the comments, it will be essential to address these points:

1) From Reviewer #2: Silencing YTHDF1, but not 2 or 3, enhanced RT in T cells. In contrast, silencing YTHDF 3, but not 1 or 2, enhanced gag mRNA expression. Silencing each in HeLa cells strongly enhanced both RT and gag mRNA expression. This needs to be laid out more clearly, and an explanation needs to be presented. The explanation given "One possibility is that different levels of YTHDF proteins in overexpression or knockdown experiments using different cell types." is confusing.

2) We need better explanation of the exact sites and population states of the modifications. What exactly do the data allow us to know? This is laid out in Reviewer #3 comment 1.

3) The cell-type specificity of the results is not adequately spelled out. If the authors feel this is a major conclusion, we need to know what is happening in relevant cells (T cells).

4) The differential effects of the various YTHDF proteins reported here are not in agreement with effects seen by others. This and other discrepancies need to be addressed or explained.

5) We need to know if the mechanism of action is thought to be RNA degradation or just inhibition of RT.

We suppose that it will not be simple to address these issues without considerable new effort, and so we want to make it clear that there is no certain path that will assure satisfaction of the reviewers. The key issues raised by Reviewer #3 will need to be addressed.

Reviewer #1:

This paper reports the effects of KD or overexpression of the three YTHDF proteins on HIV-1 replication. These are "readers" of m^6^A modification of RNAs.

This is a very active area of investigation right now. The existence of the modification on HIV-1 and many other viral RNAs has been known for some time. Recently a paper has appeared that shows that the "writers" of the modification (METTL3 and METTL14 and WTAP) and the "erasers" (the FTO proteins) have effects on HIV-1 consistent with the idea that the modifications are mostly negative. There is, however, an m^6^A on the RRE that seems to be stimulatory.

This present work maps the modifications on the viral genome and shows that a family of m^6^A binding proteins play some role in HIV-1 replication. They bind to the RNA in a characteristic pattern. They seem to be inhibitory, both during RT and during expression.

One weakness here is what they do – i.e., whether they are really "readers". Do they just interfere with RT or the ribosome? Do they do something active?

Another is that the KD effects are in general modest – often 2-5 fold (Figure 3). The KD did give 5-12x effects in one assay (Figure 2). The overexpression has bigger effects but it's not clear how to interpret these results given the abnormal levels.

The major new finding is the involvement of the YTHDF proteins, rounding out the earlier work implicating the writers and erasers. I wish we knew more about what they do. I wish we understood more about the times when m^6^A is stimulatory and inhibitory. But there is new information here, even if we cannot say that this paper presents a breakthrough in our understanding.

Reviewer #2:

The article entitled, "N6-methyladenosine of HIV-1 RNA Regulates Viral infection and HIV-1 Gag Protein Expression" by Tirumuru and colleagues is a well written and very interesting investigation into the role of N6-methyladenosine in the viral life cycle. The authors find that this modification of viral RNA has both positive and negative effects; it is inhibitory to reverse transcription due to binding by YTHDF1-3 proteins. The authors also confirm prior studies indicating that N6-methyladenosine incorporation is necessary for maximal HIV protein expression. Thus, N6-methyladenosine has both positive and negative effects depending on the stage of the viral life cycle. Overall, the experiments were well-controlled and interpreted accurately. In particular, the inhibitory effects of YTHDF1-3 on reverse transcription and viral gene expression in HeLa cells in Figure 4 and Figure 5 are quite striking. The authors also elegantly determine the binding sites of YTHDF1-3 proteins on the HIV genome and demonstrate that these sites partially correspond with sites containing N6-methyladenosine. Interestingly, YTHDF1 had a unique binding site within the *rev* gene, suggesting it plays a unique role. The data are significant and important to our understanding of this poorly understood RNA modification and its role in HIV replication.

Reviewer #3:

Overall this is an interesting story about m^6^A and HIV. A recent paper in Nature Microbiology by Rana et al. showed m^6^A promotes viral replication and via promoting REV/RRE interactions and mapped m^6^A in HIV and in the host cellular RNA. The present manuscript does not significantly advance the field in light of the prior study, and the relevance of the mechanisms seem somewhat unclear. Lastly, some of the experiments seem poorly executed.

1) Concerns with mapping m^6^A: Mapping of m^6^A sites on the HIV-1 RNA was performed in HEK293T cells using m^6^A-seq and in Jurkat cells using m^6^A -CLIP-seq. However, m^6^A mapping was never performed in the natural target cell (primary CD4^+^ T cells) of HIV-1 virus. Moreover, the resolution of these methods seems limited-it doesn't appear that the m^6^A -CLIP-Seq identifies the specific adenosine that is methylated, but regions that might contain an m^6^A. (Or perhaps the authors could show the residues that are methylated in an inset?) This is especially important for Figure 1 which look like jagged noise-how do we know these are caused by m^6^A?

The authors claim that there are 4 m^6^A's based on MS. This assumes that each site is methylated in every viral RNA. This may not be the case. The number of sites should be predicted from the mapping data, not the MS data.

2) In Figure 1 (bottom panel), the m^6^A peak profile in HEK293T and m^6^A-CLIP-seq in Jurkat cells appear strikingly different. This difference clearly suggests cell-to-cell variation in m^6^A distribution on the HIV-1 RNA. Therefore, an m^6^A map from primary CD4^+^ T cells which are the natural targets of HIV-1 virus is necessary.

3) The authors fail to identify any robust m^6^A peak/site over the input RNA in the *gag* gene (Figure 1, bottom panel), but they see an effect of overexpression of YTHDF proteins on the expression of HIV-1 Gag protein (Figure 2). At the same time, in Figure 6, several YTHDF binding sites were identified in the *gag* gene. This discrepancy is not addressed.

4) YTHDF1 and YTHDF2 have different effects on cellular mRNA. YTHDF2 (Wang et al., Nature, 2014, doi:10.1038/nature12730) mediates m^6^A-dependent mRNA degradation and YTHDF1 (Wang et al., Cell, 2015, DOI: http://dx.doi.org/10.1016/j.cell.2015.05.014) promotes mRNA translation. However, in the present manuscript, the authors demonstrate that YTHDF 1-3 show a similar effect on Gag protein expression. Overexpression of YTHDF proteins leads to a decreased expression of Gag protein (Figure 2). In Figure 5, the authors demonstrate the effects of YTHDF protein overexpression and knockdown on gag mRNA expression. Similar to the decrease in Gag protein expression, gag mRNA expression decreases upon the overexpression of YTHDF proteins. Furthermore, a shRNA-mediated knockdown leads to an increase in gag mRNA levels. From their previous findings, YTHDF1 and YTHDF2 should exhibit opposing effects on gag mRNA and protein levels. This contradictory data is not addressed. It is also possible that the changes in gag expression is due to some secondary effects of YTHDF knockdown and overexpression.

5) According to the Rana paper, HIV-1 infection modulates host gene expression and m^6^A levels in the encoded mRNAs. Protein encoded by these mRNAs (19 out of 56) were previously identified as regulators of HIV replication. Some of these proteins also directly interact with HIV proteins. These m^6^A-containing host RNAs could also be targets of YTHDF proteins. Therefore, the role of YTHDF interaction with the viral genome and inhibition of viral reverse transcription is not convincing. In cells, HIV-1 RNA can be covered with hundreds of proteins before and during reverse transcription. Why would the presence of YTHDF protein on HIV-1 genomic RNA hinder reverse transcription? It is highly likely that recruitment of YTHDF proteins induce HIV-1 RNA degradation or function in an indirect manner to affect HIV. Little mechanistic insight is provided.

6) It seems arbitrary that the authors only look at FTO and not the other m^6^A demethylase ALKBH5, yet incorporate ALKBH5 in their model.

7) The PAR-CLIP data should be aligned with the m^6^A site data so that we can see if these YTH proteins are bound at m^6^A sites. It seems like all m^6^A-Seq and PAR-CLIP experiments were done without replicates. This is a problem.

---

## [Author Response]

The reviewers were collectively in agreement that the topic of RNA m^6^A modification is of much current interest, and that there are interesting observations on its role in HIV-1 replication in the paper. But there were significant weaknesses in the data and the presentation, as should be apparent from the reviews. Some of the findings are confusing or hard to understand. Major revisions are required to answer the criticisms.

We thank reviewers for their careful reviews and helpful suggestions to improve our work. We also appreciate the editors’ clear summary of the essential points that we need to address. We have performed new experiments to address these questions and carefully revised the manuscript accordingly.

*Of the comments, it will be essential to address these points:*

1) From Reviewer #2: Silencing YTHDF1, but not 2 or 3, enhanced RT in T cells. In contrast, silencing YTHDF 3, but not 1 or 2, enhanced gag mRNA expression. Silencing each in HeLa cells strongly enhanced both RT and gag mRNA expression. This needs to be laid out more clearly, and an explanation needs to be presented. The explanation given "One possibility is that different levels of YTHDF proteins in overexpression or knockdown experiments using different cell types." is confusing.

We appreciate the suggestion from the reviewer. We have performed additional experiments to further confirm our results of the late RT and gag mRNA expression. The inhibition of RT products by YTHDF1 occurred before gag mRNA expression and is the main mechanism. Thus, to avoid confusion, we moved the gag mRNA results to Figure 4—figure supplement 1 and provided further explanations in subsection “YTHDF1–3 proteins inhibit HIV-1 infection by blocking viral reverse transcription”. We have also deleted the confusing sentence as the reviewer indicated.

*2) We need better explanation of the exact sites and population states of the modifications. What exactly do the data allow us to know? This is laid out in Reviewer #3 comment 1.*

We agree. As Reviewer #3 suggested, we have performed additional experiments to map the m^6^A sites on the HIV-1 RNAs from Jurkat T-cells and primary CD4^+^ T-cells infected with HIV-1. The new m^6^A mapping results are consistent in both cell types (Figure 1). Please refer to the details in response Reviewer #3 comment 1.

*3) The cell-type specificity of the results is not adequately spelled out. If the authors feel this is a major conclusion, we need to know what is happening in relevant cells (T cells).*

We agree. We performed additional experiments using Jurkat T-cells and primary CD4^+^ T-cells infected with replication-competent HIV-1 and obtained consistent results (Figure 1, Figure 1—figure supplement 3 and Figure 1—figure supplement 4). We also provided explanations to address cell-type specificity when we compared the results obtained from different cell types.

4) The differential effects of the various YTHDF proteins reported here are not in agreement with effects seen by others. This and other discrepancies need to be addressed or explained.

We have provided careful explanations regarding the discrepancies between previously published results and our data (see Discussion section). Please also refer to details in the following specific responses. Of note, neither recent studies (Lichinchi et al., Nature Microbiology 2016; Kennedy et al., Cell Host Microbe 2016) examined m^6^A modification of HIV-1 RNA and its effect on HIV-1 replication in primary CD4^+^ T-cells, nor systemically analyzed the role of the m^6^A writers, erasers, and readers in HIV-1 replication.

*5) We need to know if the mechanism of action is thought to be RNA degradation or just inhibition of RT.*

We agree and appreciate the constructive suggestions. To better understand the mechanism of action, we performed a new experiment to examine the impact of YTHDF1–3 on HIV-1 *gag* RNA kinetics (Figure 5). Our data suggest that YTHDF1–3 proteins can degrade HIV-1 *gag* RNA in infected cells, thereby leading to inhibition of HIV-1 reverse transcription (Figure 5).

In summary, we have made our best effort to address the key issues raised by three reviewers. We performed new experiments suggested by the reviewers and revised the manuscript accordingly. We also provide detailed responses to the specific points from three reviewers.

Reviewer #1:

*This paper reports the effects of KD or overexpression of the three YTHDF proteins on HIV-1 replication. These are "readers" of m^6^A modification of RNAs. This is a very active area of investigation right now. The existence of the modification on HIV-1 and many other viral RNAs has been known for some time. Recently a paper has appeared that shows that the "writers" of the modification (METTL3 and METTL14 and WTAP) and the "erasers" (the FTO proteins) have effects on HIV-1 consistent with the idea that the modifications are mostly negative. There is, however, an m^6^A on the RRE that seems to be stimulatory.*

*This present work maps the modifications on the viral genome and shows that a family of m^6^A binding proteins play some role in HIV-1 replication. They bind to the RNA in a characteristic pattern. They seem to be inhibitory, both during RT and during expression.*

*One weakness here is what they do – i.e., whether they are really "readers". Do they just interfere with RT or the ribosome? Do they do something active?*

We understand this important question. To better understand the mechanism of action, we performed a new experiment to examine the impact of YTHDF1–3 on HIV-1 *gag* RNA kinetics. Our data suggest that YTHDF1–3 proteins can degrade HIV-1 *gag* RNA in infected cells, thereby leading to inhibition of HIV-1 reverse transcription (Figure 5).

*Another is that the KD effects are in general modest – often 2-5 fold (Figure 3). The KD did give 5-12x effects in one assay (Figure 2). The overexpression has bigger effects but it's not clear how to interpret these results given the abnormal levels.*

We understand the concerns. To address the concern, we have performed additional knockdown experiments or quantified the knockdown efficiencies (new Figure 2, Figure 3) and presented the average results of HIV-1 infection assays from at least three independent experiments (new Figure 2, Figure 3). We agree that the overexpression generally has bigger effects than the knockdown experiments. We have explained our overexpression results cautiously and corroborated the knockdown results by demonstrating opposite effects of overexpression.

*The major new finding is the involvement of the YTHDF proteins, rounding out the earlier work implicating the writers and erasers. I wish we knew more about what they do. I wish we understood more about the times when m^6^A is stimulatory and inhibitory. But there is new information here, even if we cannot say that this paper presents a breakthrough in our understanding.*

We thank this reviewer for the supportive comments. We have performed additional experiments to better understand the role and mechanisms of the m^6^A reader proteins in HIV-1 infection. We believe that the new information in our manuscript is important to the new area studying the m^6^A modification of HIV-1 RNA or other viruses.

*Reviewer #2:*

The article entitled, "N6-methyladenosine of HIV-1 RNA Regulates Viral infection and HIV-1 Gag Protein Expression" by Tirumuru and colleagues is a well written and very interesting investigation into the role of N6-methyladenosine in the viral life cycle. The authors find that this modification of viral RNA has both positive and negative effects; it is inhibitory to reverse transcription due to binding by YTHDF1-3 proteins. The authors also confirm prior studies indicating that N6-methyladenosine incorporation is necessary for maximal HIV protein expression. Thus, N6-methyladenosine has both positive and negative effects depending on the stage of the viral life cycle. Overall, the experiments were well-controlled and interpreted accurately. In particular, the inhibitory effects of YTHDF1-3 on reverse transcription and viral gene expression in HeLa cells in Figure 4 and Figure 5 are quite striking. The authors also elegantly determine the binding sites of YTHDF1-3 proteins on the HIV genome and demonstrate that these sites partially correspond with sites containing N6-methyladenosine. Interestingly, YTHDF1 had a unique binding site within the rev gene, suggesting it plays a unique role. The data are significant and important to our understanding of this poorly understood RNA modification and its role in HIV replication.

We thank this reviewer for the positive comments and constructive suggestions. We have performed additional experiments and carefully revised the manuscript based on the reviewer’s suggestions.

*Reviewer #3:*

Overall this is an interesting story about m^6^A and HIV. A recent paper in Nature Microbiology by Rana et al. showed m^6^A promotes viral replication and via promoting REV/RRE interactions and mapped m^6^A in HIV and in the host cellular RNA. The present manuscript does not significantly advance the field in light of the prior study, and the relevance of the mechanisms seem somewhat unclear. Lastly, some of the experiments seem poorly executed.

We thank this reviewer for the careful review and constructive suggestions to improve our work and manuscript. We have performed additional experiments as the reviewer suggested and carefully revised our manuscript accordingly. We are confident that our revised manuscript can provide new information to advance this new field studying HIV-1 RNA methylation.

*1) Concerns with mapping m^6^A: Mapping of m^6^A sites on the HIV-1 RNA was performed in HEK293T cells using m^6^A-seq and in Jurkat cells using m^6^A-CLIP-seq. However, m^6^A mapping was never performed in the natural target cell (primary CD4^+^ T cells) of HIV-1 virus. Moreover, the resolution of these methods seems limited-it doesn't appear that the m^6^A-CLIP-Seq identifies the specific adenosine that is methylated, but regions that might contain a m^6^A. (Or perhaps the authors could show the residues that are methylated in an inset?) This is especially important for Figure 1 which look like jagged noise-how do we know these are caused by m^6^A?*

*The authors claim that there are 4 m^6^A's based on MS. This assumes that each site is methylated in every viral RNA. This may not be the case. The number of sites should be predicted from the mapping data, not the MS data.*

We understand the reviewer’s concerns and appreciate the helpful suggestions. As the reviewer suggested, we have performed additional experiments to map the m^6^A sites on the HIV-1 RNAs from Jurkat T-cells and primary CD4^+^ T-cells infected with a replication-competent HIV-1. The new m^6^A mapping results are consistent in both cell types (Figure 1). To avoid potential confusion, we have moved the previous m^6^A mapping results of HIV-1 RNA from HEK293T cells to Figure 1—figure supplement 1.

To identify the specific adenosine on the HIV-1 RNA that is m^6^A-methylated, we have planned to employ a photo-crosslinking-assisted m^6^A sequencing strategy that more accurately defines sites of modification (Chen, K et al. *Angew Chem Int Ed Engl* 2015), and validate the result with a base-resolution method termed SCARLET (site-specific cleavage and radioactive labeling followed by ligation-assisted extraction and thin-layer chromatography) (Liu N. et al. RNA, 2013). However, we need to optimize the protocols of both methods to define the specific m^6^A residues on the HIV-1 RNA, which would require additional controls and time to complete. We feel that this experiment is beyond the current scope and focus of our study, which we will complete in the future.

To investigate whether purified HIV-1 virion RNA contains m^6^A, we quantified HIV-1 RNA m^6^A level using liquid chromatography-mass spectrometry (Figure 1—figure supplement 2). This is a novel approach because none of two recent studies (Lichinchi et al., *Nature Microbiology* 2016; Kennedy et al., *Cell Host Microbe* 2016) directly measured m^6^A level of HIV-1 RNA from purified HIV-1 particles. Considering 35.8% of HIV-1 genomic RNA (9,173 nucleotides) sequence are adenosines (van Hemert et al., *Virus Res.* 2014), based on our data, we estimated 3-4 sites of the m^6^A modification in each copy of HIV-1 gRNA, which match the numbers of m^6^A peaks identified by m^6^A-seq (Figure 1 and Figure 1—figure supplement 1). We believe that this new information is helpful to further define the specific m^6^A residues on the HIV-1 RNA and study their functions and mechanisms.

2) In Figure 1 (bottom panel), the m^6^A peak profile in HEK293T and m^6^A-CLIP-seq in Jurkat cells appear strikingly different. This difference clearly suggests cell-to-cell variation in m^6^A distribution on the HIV-1 RNA. Therefore, a m^6^A map from primary CD4^+^ T cells which are the natural targets of HIV-1 virus is necessary.

We agree. As the reviewer suggested, we have performed additional experiments to map the m^6^A sites on the HIV-1 RNAs from Jurkat T-cells and primary CD4^+^ T-cells infected with a replication-competent HIV-1. The new m^6^A mapping results are consistent in both cell types (Figure 1).

3) The authors fail to identify any robust m^6^A peak/site over the input RNA in the gag gene (Figure 1, bottom panel), but they see an effect of overexpression of YTHDF proteins on the expression of HIV-1 Gag protein (Figure 2). At the same time, in Figure 6, several YTHDF binding sites were identified in the gag gene. This discrepancy is not addressed.

Thanks for the comments and we agree. We now provided new results of HIV-1 RNA m^6^A mapping in both Jurkat T-cells and primary CD4^+^ T-cells (Figure 1), which show two m^6^A peaks in the *gag* gene. Together, these results confirm the m^6^A modification of HIV-1 RNA despite some differences in m^6^A distributions in HIV-1 infected CD4^+^ T-cells compared to transfected HEK293T cells.

4) YTHDF1 and YTHDF2 have different effects on cellular mRNA. YTHDF2 (Wang et al., Nature, 2014, doi:10.1038/nature12730) mediates m^6^A-dependent mRNA degradation and YTHDF1 (Wang et al., Cell, 2015, DOI: http://dx.doi.org/10.1016/j.cell.2015.05.014) promotes mRNA translation. However, in the present manuscript, the authors demonstrate that YTHDF 1-3 show a similar effect on Gag protein expression. Overexpression of YTHDF proteins leads to a decreased expression of Gag protein (Figure 2). In Figure 5, the authors demonstrate the effects of YTHDF protein overexpression and knockdown on gag mRNA expression. Similar to the decrease in Gag protein expression, gag mRNA expression decreases upon the overexpression of YTHDF proteins. Furthermore, a shRNA-mediated knockdown leads to an increase in gag mRNA levels. From their previous findings, YTHDF1 and YTHDF2 should exhibit opposing effects on gag mRNA and protein levels. This contradictory data is not addressed. It is also possible that the changes in gag expression is due to some secondary effects of YTHDF knockdown and overexpression.

We agree and appreciate the comments. We have added discussions on different effects of YTHDF1 and YTHDF2 on cellular mRNA and translation (Discussion section, second paragraph). Indeed, we observed a unique high peak within the *rev* gene of HIV-1 RNA bound by YTHDF1 (Figure 1), while YTHDF1 and YTHDF2 appear to have similar inhibitory effects on HIV-1 infection. It is possible that YTHDF1 and YTHDF2 may interact with different host proteins that directly or indirectly affect HIV-1 replication and lead to similar effects on viral inhibition.

5) According to the Rana paper, HIV-1 infection modulates host gene expression and m^6^A levels in the encoded mRNAs. Protein encoded by these mRNAs (19 out of 56) were previously identified as regulators of HIV replication. Some of these proteins also directly interact with HIV proteins. These m^6^A-containing host RNAs could also be targets of YTHDF proteins. Therefore, the role of YTHDF interaction with the viral genome and inhibition of viral reverse transcription is not convincing. In cells, HIV-1 RNA can be covered with hundreds of proteins before and during reverse transcription. Why would the presence of YTHDF protein on HIV-1 genomic RNA hinder reverse transcription? It is highly likely that recruitment of YTHDF proteins induce HIV-1 RNA degradation or function in an indirect manner to affect HIV. Little mechanistic insight is provided.

We agree and appreciate the constructive suggestions. To better understand the mechanism of action of YTHDF proteins, we have performed a new experiment to examine the impact of YTHDF1–3 on HIV-1 *gag* RNA kinetics (Figure 5). Our data indicate that YTHDF1–3 proteins can degrade HIV-1 *gag* RNA in infected cells, thereby leading to inhibition of HIV-1 reverse transcription (Figure 5).

6) It seems arbitrary that the authors only look at FTO and not the other m^6^A demethylase ALKBH5, yet incorporate ALKBH5 in their model.

We agree. We have added the result of knockdown ALKBH5 alone and double knockdown of FTO and ALKBH5 in new Figure 6.

*7) The PAR-CLIP data should be aligned with the m^6^A site data so that we can see if these YTH proteins are bound at m^6^A sites. It seems like all m^6^A-Seq and PAR-CLIP experiments were done without replicates. This is a problem.*

We agree. We have aligned the CLIP-seq data (Figure 1) with the m^6^A site mapping data (Figure 1) in new Figure 1. We have also repeated the m^6^A-Seq experiments using both Jurkat cells and primary CD4^+^ T-cells. We identified similar profiles of m^6^A peaks in HIV-1 RNA from these two cell types, which are mainly enriched in the 5' and 3' UTRs, the *rev* and *gag* genes of the HIV-1 genome (Figure 1).